# FedP3: Federated Personalized and Privacy-friendly Network Pruning under Model Heterogeneity

**Kai Yi**[1][*]**, Nidham Gazagnadou**[2]**, Peter Richtárik**[1]**, Lingjuan Lyu**[2]
[1]KAUST, [2]Sony AI
`kai.yi@kaust.edu.sa, nidham.gazagnadou@sony.com`
`peter.richtarik@kaust.edu.sa, lingjuan.lv@sony.com`

## Abstract

The interest in federated learning has surged in recent research due to its unique ability to train a global model using privacy-secured information held locally on each client. This paper pays particular attention to the issue of client-side model heterogeneity, a pervasive challenge in the practical implementation of FL that escalates its complexity. Assuming a scenario where each client possesses varied memory storage, processing capabilities and network bandwidth - a phenomenon referred to as system heterogeneity - there is a pressing need to customize a unique model for each client. In response to this, we present an effective and adaptable federated framework FedP3, representing **Fed**erated **P**ersonalized and **P**rivacy-friendly network **P**runing, tailored for model heterogeneity scenarios. Our proposed methodology can incorporate and adapt well-established techniques to its specific instances. We offer a theoretical interpretation of FedP3 and its locally differential-private variant, DP-FedP3, and theoretically validate their efficiencies.

## 1 Introduction

Federated learning (FL) (McMahan et al., 2017; Konečný et al., 2016) has emerged as a significant machine learning paradigm wherein multiple clients perform computations on their private data locally and subsequently communicate their findings to a remote server. Standard FL can be articulated as an optimization problem, specifically the Empirical Risk Minimization (ERM) given by

$$\min_{W \in \mathbb{R}^d} f(W) \coloneqq \frac{1}{n} \sum_{i=1}^{n} f_i(W) \ , \tag{1}$$

where $W$ represents the shared global network parameters, $f_i$ denotes the local objective for client $i$, and $n$ is the total number of clients.

Distinguishing it from conventional distributed learning, FL predominantly addresses heterogeneity stemming from both data and model aspects. Data heterogeneity characterizes the fact that the local data distribution across clients can vary widely. Such variation is rooted in real-world scenarios where clients or users exhibit marked differences in their data, reflective of the variety of sensors or software Jiang et al. (2020), of users' unique preferences, etc. Li et al. (2020a). Recent works Zhao et al. (2018) showed how detrimental the non-iidness of the local data could be on the training of a FL model. This phenomenon known as client-drift, is intensively studied to develop methods limiting its impact on the performance (Karimireddy et al., 2020; Gao et al., 2022b; Mendieta et al., 2022).

Furthermore, given disparities among clients in device resources, e.g., energy consumption, computational capacities, memory storage or network bandwidths, model heterogeneity becomes a pivotal consideration. To avoid restricting the global model's architecture to the largest that is compatible with all clients, recent methods aim at reducing its size differently for each client to extract the utmost of their capacities. This can be referred to as constraint-based local model personalization (Gao

---

[*]Work done during an internship at Sony AI.

et al., 2022a). In such a context, clients often train a pruned version of the global model (Jiang et al., 2022b; Diao et al., 2021) before transmitting it to the server for aggregation (Li et al., 2021). A contemporary and influential offshoot of this is Independent Subnetwork Training (IST) (Yuan et al., 2022). It hinges on the concept that each client trains a subset of the main server-side model, subsequently forwarding the pruned model to the server. Such an approach significantly trims local computational burdens in FL (Dun et al., 2023).

Our research, while aligning with the IST premise, brings to light some key distinctions. A significant observation from our study is the potential privacy implications of continuously sending the complete model back to the server. Presently, even pruned networks tend to preserve the overarching structure of the global model. In this paper, we present an innovative approach to privacy-friendly pruning. Our method involves transmitting only select segments of the global model back to the server. This technique effectively conceals the true structure of the global model, thus achieving a delicate balance between utility and confidentiality. As highlighted in Zeiler & Fergus (2014), different layers within networks demonstrate varied capacities for representation and semantic interpretation. The challenge of securely transferring knowledge from client to server, particularly amidst notable model heterogeneity, is an area that has not been thoroughly explored. It's pertinent to acknowledge that the concept of gradient pruning as a means of preserving privacy was initially popularized by the foundational work of Zhu et al. (2019). Following this, studies such as Huang et al. (2020) have further investigated the efficacy of DNN pruning in maintaining privacy.

Besides, large language models (LLMs) have garnered significant attention and have been applied to a plethora of real-world scenarios (Brown et al., 2020; Chowdhery et al., 2022; Touvron et al., 2023) recently. However, the parameter count of modern LLMs often reaches the billion scale, making it challenging to utilize user or client information and communicate within a FL framework. We aim to explore the feasibility of training a more compact local model and transmitting only a subset of the global network parameters to the server, while still achieving commendable performance.

From a formulation standpoint, our goal is to optimize the following objective, thereby crafting a global model under conditions of model heterogeneity:

$$\min_{W_1,\ldots,W_n \in \mathbb{R}^d} f(W) \coloneqq h\left(f_1(W_1), f_2(W_2), \ldots, f_n(W_n)\right) \ , \tag{2}$$

where $W_i$ denotes the model downloaded from client $i$ to the server, which can differ as we allow global pruning or other sparsification strategies. The global model $W$ is a function of $\{W_1, W_2, \ldots, W_n\}$, $f_i$ the local objective for client $i$ and $n$ the total number of clients. Function $h$ is the aggregation mapping from the clients to the server. In conventional FL, it's assumed that function $h$ is the average and all $W_1 = \ldots W_n = W$, which means the full global model is downloaded from the server to every client. When maintaining a global model $W$, this gives us $f(x) \coloneqq \frac{1}{n}\sum_{i=1}^{n} f_i(W)$, which aligns with the standard empirical risk minimization (ERM).

In this paper, we introduce an efficient and adaptable federated network pruning framework tailored to address model heterogeneity. The main contributions of our framework, denoted as FedP3 (**Fed**erated **P**ersonalized and **P**rivacy-friendly network **P**runing) algorithm, are:

- **Versatile Framework:** Our framework allows personalization based on each client's unique constraints (computational, memory, and communication).

- **Dual-Pruning Method:** Incorporates both global (server to client) and local (client-specific) pruning strategies for enhanced efficiency.

- **Privacy-Friendly Approach:** Ensures privacy-friendly to each client by limiting the data shared with the server to only select layers post-local training.

- **Managing Heterogeneity:** Effectively tackles data and model diversity, supporting non-iid data distributions and various client-model architectures.

- **Theoretical Interpretation:** Provides a comprehensive analysis of global pruning and personalized model aggregation. Discusses convergence theories, communication costs, and the advantages over existing methodologies.

- **Local Differential-Privacy Algorithm:** Introduces LDP-FedP3, a novel local differential privacy algorithm. Outlines privacy guarantees, utility, and communication efficiency.

---

**Algorithm 1** FedP3

---

1: **Input:** Client $i$ has data $X_i$ for $i \in [n]$, the number of local updates $K$, the number of communication rounds $T$, initial model weights $W_t = \{W_t^0, W_t^1, \dots, W_t^L\}$ on the server for $t = 0$
2: Server specifies the server pruning mechanism $P_i$, the client pruning mechanism $Q_i$, and the set of layers to train $L_i \subseteq [L]$ for each client $i \in [n]$
3: **for** $t = 0, 1, \dots, T - 1$ **do**
4:      Server samples a subset of participating clients $\mathcal{C}_t \subset [n]$
5:      Server sends the layer weights $W_t^l$ for $l \in L_i$ to client $i \in \mathcal{C}_t$ for training
6:      Server sends the pruned weights $P_i \odot W_t^l$ for $l \notin L_i$ to client $i \in \mathcal{C}_t$
7:      **for** each client $i \in \mathcal{C}_t$ in parallel **do**
8:          Initialize $W_{t,0}^l = W_t^l$ for all $l \in [L_i]$ and $W_{t,0}^l = P_i \odot W_t^l$ for all $l \notin [L_i]$
9:          **for** $k = 0, 1, \dots, K - 1$ **do**
10:             Compute $W_{t,k+1} \leftarrow \texttt{LocalUpdate}(W_{t,k}, X_i, L_i, Q_i, k)$,
                where $W_{t,k} := \{W_{t,k}^0, W_{t,k}^1, \dots, W_{t,k}^L\}$
11:          **end for**
12:          Send $\cup_{l \in L_i} W_{t,K}^l$ to the server
13:      **end for**
14:      Server aggregates $W_{t+1} = \texttt{Aggregation}(\cup_{i \in [n]} \cup_{l \in L_i} W_{t,K}^l)$
15: **end for**
16: **Output:** $W_T$

---

## 2 APPROACH

We focus on the training of neural networks within the FL paradigm. Consider a global model

$$W := \{W^0, W^1, \dots, W^L, W^{\text{out}}\} \ ,$$

where $W^0$ represents the weights of the input layer, $W^{\text{out}}$ the weights of the final output layer, and $L$ the number of hidden layers. Each $W^l$, for all $l \in \mathcal{L} := \{0, 1, \dots, L\}$, denotes the model parameters for layer $l$. We distribute the complete dataset $X$ across $n$ clients following a specific distribution, which can be non-iid. Each client then conducts local training on its local data denoted by $X_i$.

**Algorithmic overview.** In Algorithm 1, we introduce the details of our proposed general framework called **Fed**erated **P**ersonalized and **P**rivacy-friendly network **P**runing (FedP3). For every client $i \in [n]$, we assign predefined pruning mechanisms $P_i$ and $Q_i$, determined by the client's computational capacity and network bandwidth (see Line 2). Here, $P_i$ denotes the maximum capacity of a pruned global model $W$ sent to client $i$, signifying server-client global pruning. On the other hand, $Q_i$ stands for the local pruning mechanism, enhancing both the speed of local computation and the robustness (allowing more dynamics) of local network training.

In Line 4, we opt for partial client participation by selecting a subset of clients $\mathcal{C}_t$ from the total pool $[n]$. Unlike the independent subnetwork training approach, Lines 5–6 employ a personalized server-client pruning strategy. This aligns with the concept of collaborative training. Under this approach, we envision each client learning a subset of layers, sticking to smaller neural network architectures of the global model. Due to the efficient and privacy-friendly communication, such a method is not only practical but also paves a promising path for future research in FL-type training and large language models.

The server chooses a layer subset $L_i$ for client $i$ and dispatches the pruned weights, conditioned by $P_i$, for the remaining layers. Local training spans $K$ steps (Lines 8–12), detailed in Algorithm 2. To uphold a privacy-friendly framework, only weights $\cup_{l \in L_i} W_{t,K}^l$ necessary for training of each client $i$ are transmitted to the server (Line 12). The server concludes by aggregating the weights received from every client to forge the updated model $W_{t+1}$, as described in Algorithm 3. We also provide an intuitive pipeline in Figure 1.

**Local update.** Our proposed framework, FedP3, incorporates dynamic network pruning. In addition to personalized task assignments for each client $i$, our local update mechanism supports diverse pruning strategies. Although efficient pruning strategies in FL remain an active research

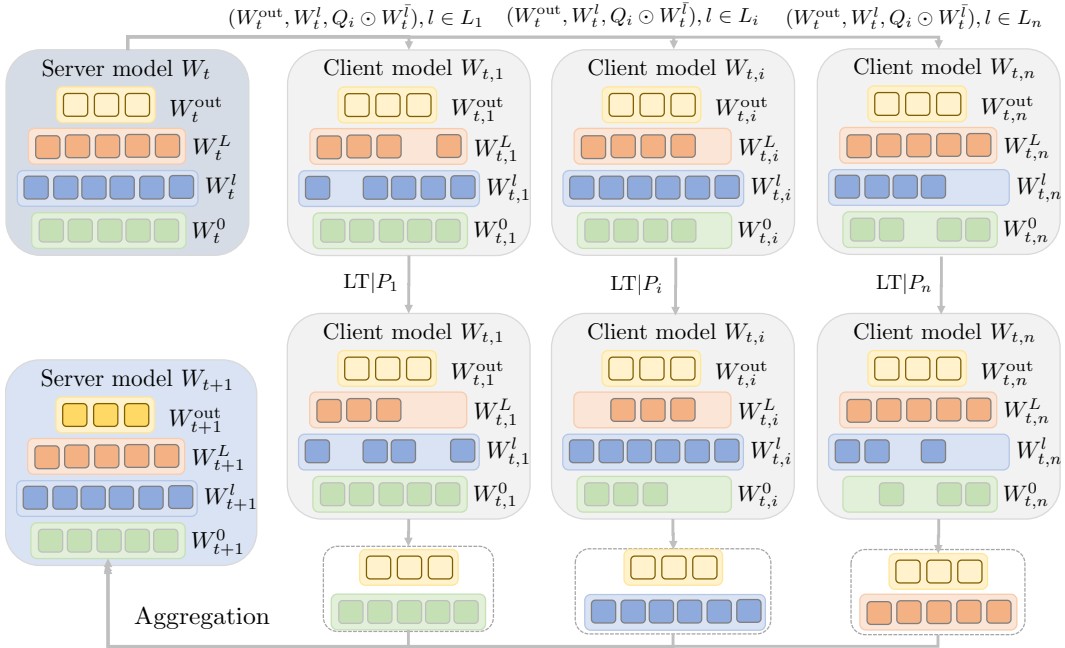

Figure 1: Pipeline illustration of our proposed framework FedP3.

---

**Algorithm 2** `LocalUpdate`

1: **Input:** $W_{t,k}, X_i, L_i, Q_i, k$
2: Generate the step-wise local pruning ratio $q_{i,k}$ conditioned on $P_i$ and $Q_i$
3: Local training $\left(\cup_{l \in L_i} W_{t,k}^l\right) \cup \left(\cup_{l \notin L_i} q_{i,k} \odot P_i \odot W_t^l\right)$ using local data $X_i$
4: **Output:** $W_{t,k+1}$

---

area (Horváth et al., 2021; Alam et al., 2022; Liao et al., 2023), we aim to determine if our framework can accommodate various strategies and yield significant insights. In this context, we examine different local update rules as described in Algorithm 2. We evaluate three distinct strategies: *fixed without pruning*, *uniform pruning*, and *uniform ordered dropout*.

Assuming our current focus is on $W_{t,k}^l$, where $l \notin L_i$, after procuring the pruned model conditioned on $P_i$ from the server, we denote the sparse model we obtain by $P_i \odot W_{t,0}^l$. Here:

- *Fixed without pruning* implies that we conduct multiple steps of the local update without additional local pruning, resulting in $P_i \odot W_{t,K}^l$.

- *Uniform pruning* dictates that for every local iteration $k$, we randomly generate the probability $q_{i,k}$ and train the model $q_{i,k} \odot P_i \odot W_{t,K}^l$.

- *Uniform ordered dropout* is inspired by Horváth et al. (2021). In essence, if $P_i \odot W_{t,0}^l \in \mathbb{R}^{d_1 \times d_2}$ (extendable to 4D convolutional weights; however, we reference 2D fully connected layer weights here), we retain only the subset $P_i \odot W_{t,0}^l[: q_{i,k}d_1, : q_{i,k}d_2]$ for training purposes. $[: q_{i,k}d_1]$ represents we select the first $q_{i,k} \times d_1$ elements from the total $d_1$ elements.

Regardless of the chosen method, the locally deployed model is given by $\left(\cup_{l \in L_i} W_{t,k}^l\right) \cup \left(\cup_{l \notin L_i} q_{i,k} \odot P_i \odot W_{t,k}^l\right)$, as highlighted in Algorithm 2 Line 3.

---

**Algorithm 3** `Aggregation`

---

1: **Input:** $\cup_{i \in [n]} \cup_{l \in L_i} W_{t,K}^l$
2: *Simple Averaging:*
3:     $W_{t+1}^l \leftarrow \mathrm{Avg}\left(W_{t,K,i}^l\right)$ for all nodes with $l \in L_i$
4: *Weighted Averaging:*
5:     Construct the aggregation weighting $\alpha_i$ for each client $i$
6:     $W_{t+1}^l \leftarrow \mathrm{Avg}\left(\alpha_i W_{t,K,i}^l\right)$ for all nodes with $l \in L_i$
7: *Attention Averaging:*
8:     Construct an attention mapping layer annotated by function $h$
9:     $W_{t+1}^l \leftarrow h\left(W_{t,K,i}^l\right)$ for all nodes with $l \in L_i$
10: **Output:** $W_{t+1}$

---

**Layer-wise aggregation.**    Our Algorithm 1 distinctively deviates from existing methods in Line 12 as each client forwards only a portion of information to the server, thus prompting an investigation into optimal aggregation techniques. In Algorithm 3 we evaluate three aggregation methodologies:

- *Simple averaging* computes the mean of all client contributions that include a specific layer $l$. This option is presented in Line 3.

- *Weighted averaging* adopts a weighting scheme based on the number of layers client $i$ is designated to train. Specifically, the weight for aggregating $W_{t,K,i}^l$ from client $i$ is given by $|L_i| / \sum_{j=1}^n |L_j|$, analogous to importance sampling. This option is presented in Line 5

- *Attention-based averaging* introduces an adaptive mechanism where an attention layer is learned specifically for layer-wise aggregation. This option is presented in Line 9.

## 3 THEORETICAL ANALYSIS

Our work refines independent subnetwork training (IST) by adding personalization and layer-level sampling, areas yet to be fully explored (see Appendix A.2 for related work). Drawing on the sketch-based analysis from Shulgin & Richtárik (2023), we aim to thoroughly analyze FedP3, enhancing the sketch-type design concept in both scope and depth.

Consider a global model denoted as $w \in \mathbb{R}^d$. In Shulgin & Richtárik (2023), a sketch $\mathcal{C}_i^k \in \mathbb{R}^{d \times d}$ represents submodel computations by weights permutations. We extend this idea to a more general case encompassing both global pruning, denoted as $\mathbf{P} \in \mathbb{R}^{d \times d}$, and personalized model aggregations, denoted as $\mathbf{S} \in \mathbb{R}^{d \times d}$. Now we first present the formal definitions.

**Definition 1** (Global Pruning Sketch $\mathbf{P}$). *Let a random subset $\mathcal{S}$ of $[d]$ is a proper sampling such that the probability $c_j := \mathrm{Prob}(j \in S) > 0$ for all $j \in [d]$. Then the biased diagonal sketch with $\mathcal{S}$ is $\mathbf{P} := \mathrm{Diag}(p_s^1, p_s^2, \cdots, p_s^d)$, where $p_s^j = 1$ if $j \in S$ otherwise 0.*

Unlike Shulgin & Richtárik (2023), we assume client-specific sampling with potential weight overlap. For simplicity, we consider all layers pruned from the server to the client, a more challenging case than the partial pruning in FedP3 (Algorithm 1). The convergence analysis of this global pruning sketch is in Appendix C.4.

**Definition 2** (Personalized Model Aggregation Sketch $\mathbf{S}$). *Assume $d \geq n$, $d = sn$, where $s \geq 1$ is an integer. Let $\pi = (\pi_1, \cdots, \pi_d)$ be a random permutation of the set $[d]$. The number of parameters per layer $n_l$, assume $s$ can be divided by $n_l$. Then, for all $x \in \mathbb{R}^d$ and each $i \in [n]$, we define $\mathbf{S}$ as $\mathbf{S} := n \sum_{j=s(i-1)+1}^{si} e_{\pi_j} e_{\pi_j}^\top$.*

Sketch $\mathbf{S}$ is based on the permutation compressor technique from Szlendak et al. (2021). Extending this idea to scenarios where $d$ is not divisible by $n$ follows a similar approach as outlined in Szlendak et al. (2021). To facilitate analysis, we apply a uniform parameter count $n_l$ across layers, preserving layer heterogeneity. For layers with fewer parameters than $d_L$, zero-padding ensures operational consistency. This uniform distribution assumption maintains our findings' generality and simplifies the discussion. Our method assumes $s$ divides $d_l$, streamlining layer selection over individual elements. The variable $v$ denotes the number of layers chosen per client, shaping a more analytically conducive framework for FedP3, detailed in Algorithm 4 in the Appendix.

**Theorem 1** (Personalized Model Aggregation). *Let Assumption 1 holds. Iterations $K$, choose step-size $\gamma \leq \left\{ 1/L_{\max}, 1/\sqrt{\bar{L} L_{\max} K} \right\}$. Denote $\Delta_0 := f(w^0) - f^{\inf}$. Then for any $K \geq 1$, the iterates $w^k$ of* FedP3 *in Algorithm 4 satisfy*

$$\min_{0 \leq k \leq K-1} \mathbb{E}\left[ \left\| \nabla f(w^k) \right\|^2 \right] \leq \frac{2(1 + \bar{L} L_{\max} \gamma^2)^K}{\gamma K} \Delta_0. \tag{3}$$

We have achieved a total communication cost of $\mathcal{O}\left( d/\epsilon^2 \right)$, marking a significant improvement over unpruned methods. This enhancement is particularly crucial in FL for scalable deployments, especially with a large number of clients. Our approach demonstrates a reduction in communication costs by a factor of $\mathcal{O}\left( n/\epsilon \right)$. In the deterministic setting of unpruned methods, we compute the exact gradient, in contrast to bounding the gradient as in Lemma 1. Remarkably, by applying the smoothness-based bound condition (Lemma 1) to both FedP3 and the unpruned method, we achieve a communication cost reduction by a factor of $\mathcal{O}(d/n)$ for free. This indicates that identifying a tighter upper gradient bound could potentially lead to even more substantial theoretical improvements in communication efficiency. A detailed analysis is available in Appendix C.2. We have also presented an analysis of the locally differential-private variant of FedP3, termed LDP-FedP3, in Theorem 2.

**Theorem 2** (LDP-FedP3). *Under Assumptions 1 and 2, with the use of Algorithm 5, consider the number of samples per client to be $m$ and the number of steps to be $K$. Let the local sampling probability be $q \equiv b/m$. For constants $c'$ and $c$, and for any $\epsilon < c' q^2 K$ and $\delta \in (0, 1)$,* LDP-FedP3 *achieves $(\epsilon, \delta)$-LDP with $\sigma^2 = \frac{cKC^2 \log(1/\epsilon)}{m^2 \epsilon^2}$.*

*Set $K = \max\left\{ \frac{m\epsilon\sqrt{L\Delta_0}}{C\sqrt{cd\log(1/\delta)}}, \frac{m^2\epsilon^2}{cd\log(1/\delta)} \right\}$ and $\gamma = \min\left\{ \frac{1}{L}, \frac{\sqrt{\Delta_0 cd\log(1/\delta)}}{Cm\epsilon\sqrt{L}} \right\}$, we have:*

$$\frac{1}{K} \sum_{k=0}^{K-1} \mathbb{E}\left[ \left\| \nabla f(w^t) \right\|^2 \right] \leq \frac{2C\sqrt{Lcd\log(1/\sigma)}}{m\epsilon} = \mathcal{O}\left( \frac{C\sqrt{Ld\log(1/\delta)}}{m\epsilon} \right).$$

*Consequently, the total communication cost is:*

$$C_{\text{LDP-FedP3}} = \mathcal{O}\left( \frac{m\epsilon\sqrt{dL\Delta_0}}{C\sqrt{\log(1/\delta)}} + \frac{m^2\epsilon^2}{\log(1/\delta)} \right).$$

We establish the privacy guarantee and communication cost of LDP-FedP3. Our analysis aligns with the communication complexity in Li et al. (2022) while providing a more precise convergence bound. Further details and comparisons with existing work are discussed in Appendix C.3.

## 4 EXPERIMENTS

### 4.1 DATASETS AND SPLITTING TECHNIQUES

We utilize benchmark datasets CIFAR10/100 Krizhevsky et al. (2009), a subset of EMNIST labeled EMNIST-L Cohen et al. (2017), and FashionMNIST Xiao et al. (2017), maintaining standard train/test splits as in McMahan et al. (2017) and Li et al. (2020b). While CIFAR100 has 100 labels, the others have 10, with a consistent data split of 70% for training and 30% for testing. Details on these splits are in Table 3 in the Appendix. For non-iid splits in these datasets, we employ class-wise and Dirichlet non-iid strategies, detailed in Appendix B.2.

### 4.2 OPTIMAL LAYER OVERLAPPING AMONG CLIENTS

**Datasets and Models Specifications.** In this section, our objective is to develop a communication-efficient architecture that also preserves accuracy. We conducted extensive experiments on recognized datasets like CIFAR10/100 and FashionMNIST, using a neural network with two convolutional layers (denoted as `Conv`) and four fully-connected layers (`FC`). For EMNIST-L, our model includes four `FC` layers including the output layer. This approach simplifies the identification of optimal layer overlaps among clients. We provide the details of network architectures in Appendix B.3.

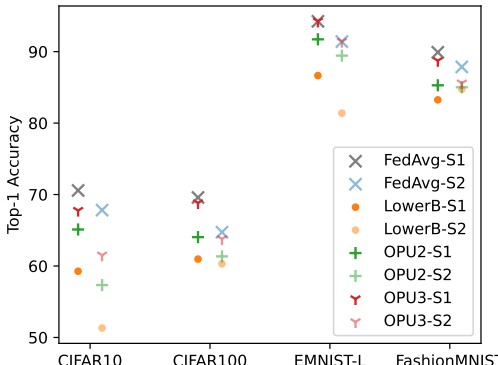 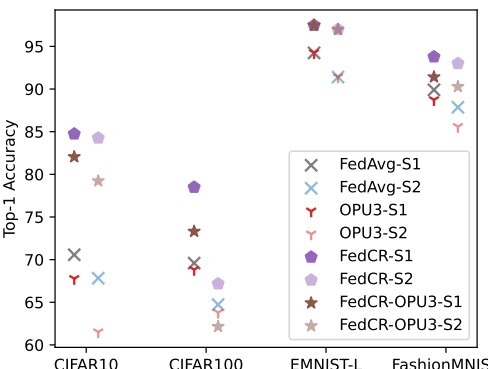

Figure 2: Comparative Analysis of Layer Overlap Strategies: The left figure presents a comparative study of different overlapping layer configurations across four major datasets. On the right, we extend this comparison to include the state-of-the-art personalized FL method, FedCR. In this context, S1 refers to a class-wise non-iid distribution, while S2 indicates a Dirichlet non-iid distribution.

**Layer Overlapping Analysis.** Figure 2 presents a comparison of different layer overlapping strategies. For Optional Pruning Uniformly with selection of 2 layers (OPU2) represents the selection of two uniformly chosen layers from the entire network for training, while OPU3 involves 3 such layers. LowerB denotes the scenario where only one layer's parameters are trained per client, serving as a potential lower bound benchmark. All clients participate in training the final FC layer (denoted as FFC). "S1" and "S2" signify class-wise and Dirichlet data distributions, respectively. For example, FedAvg-S1 shows the performance of FedAvg under a class-wise non-iid setting. Given that a few layers are randomly assigned for each client to train, we assess the communication cost on average. In CIFAR10/100 and FashionMNIST training, by design, we obtain a 20% communication reduction for OPU3, 40% for OPU2, and 60% for LowerB. Remarkably, OPU3 shows comparable performance to FedAvg, with only 80% of the parameters communicated. Computational results in the Appendix B.5 (Figure 6) elucidate the outcomes of randomly sampling a single layer (LowerB). Particularly in CIFAR10, clients training on FC2+FFC layers face communication costs more than 10,815 times higher than those training on Conv1+FFC layers, indicating significant model heterogeneity.

Beyond validating FedAvg, we compare with the state-of-the-art personalized FL method FedCR Zhang et al. (2023) (details in Appendix B.4), as shown on the right of Figure 2. Our method (FedCR-OPU3), despite 20% lower communication costs, achieves promising performance with only a 2.56% drop on S1 and a 3.20% drop on S2 across four datasets. Additionally, Figure 2 highlights the performance differences between the two non-iid data distribution strategies, S1 and S2. The average performance gap across LowerB, OPU2, and OPU3 is 3.55%. This minimal reduction in performance across all datasets underscores the robustness and stability of our FedP3 pruning strategy in diverse data distributions within FL.

**Larger Network Verifications.** Our assessment extends beyond shallow networks to the more complex ResNet18 model He et al. (2016), tested with CIFAR10 and CIFAR100 datasets. Figure 3 illustrates the ResNet18 architecture, composed of four blocks, each containing four layers with skip connections, plus an input and an output layer, totaling 18 layers. A key focus of our study is to evaluate the efficiency of training this heterogeneous model using only a partial set of its layers. We performed layer ablations in blocks 2 and 3 (B2 and B3), as shown in Figure 1. The notation -B2-B3(full) indicates complete random pruning of B2 or B3, with the remaining structure sent to the server. -B2(part) refers to pruning the first or last two layers in B2. We default the global pruning ratio from server to client at 0.9, implying that the locally deployed model is approximately 10% smaller than the global model. Results in Figure 1 demonstrate that dropping random layers from ResNet18 does not significantly impact performance, sometimes even enhancing it. Compared with Full, -B2(part) and -B3(part) achieved a 6.25% reduction in communication costs with only a 1.03% average decrease in performance. Compared to the standard FedAvg without pruning, this is a 16.63% reduction, showcasing the efficiency of our FedP3 method. Remarkably, -B3(part) even surpassed the Full model in performance. Additionally, -B2-B3(full) re-

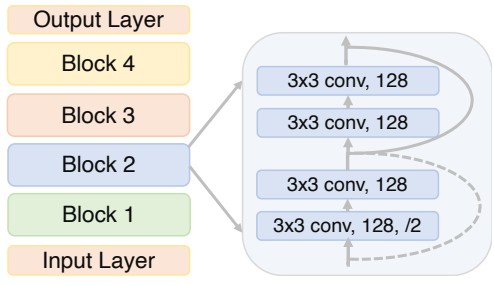

Figure 3: ResNet18 architecture.

| Method | CIFAR10 | CIFAR100 |
|---|---|---|
| Full | 73.25 | 63.33 |
| -B2-B3 (full) | 65.68 | 58.26 |
| -B2 (part) | 72.09 | 61.11 |
| -B3 (part) | 73.47 | 62.39 |

Table 1: Performance of ResNet18 under class-wise non-iid conditions. The global pruning ratio from server to client is maintained at 0.9 for all baseline comparisons by default.

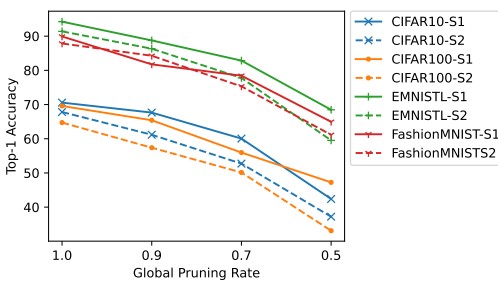 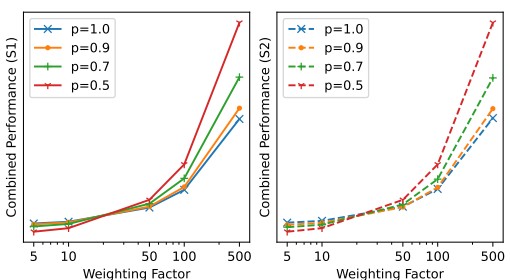

Figure 4: Comparative Analysis of Server to Client Global Pruning Strategies: The left portion displays Top-1 accuracy across four major datasets and two distinct non-IID distributions, varying with different global pruning rates. On the right, we quantitatively assess the trade-off between model size and accuracy.

sulted in a 12.5% average reduction in communication costs (21.25% less compared to unpruned FedAvg), with just a 6.32% performance drop on CIFAR10 and CIFAR100. These results demonstrate the potential of FedP3 for effective learning in LLMs.

## 4.3 KEY ABLATION STUDIES

Our framework, detailed in Algorithm 1, critically depends on the choice of pruning strategies. The FedP3 algorithm integrates both server-to-client global pruning and client-specific local pruning. Global pruning aims to minimize the size of the model deployed locally, while local pruning focuses on efficient training and enhanced robustness.

### 4.3.1 EXPLORING SERVER TO CLIENT GLOBAL PRUNING STRATEGIES

We investigate various global pruning ratios and their impacts, as shown in the left part of Figure 4. A global pruning rate of 0.9 implies the local model has 10% fewer parameters than the global model. When comparing unpruned (rate 1.0) scenarios, we note an average performance drop of 5.32% when reducing the rate to 0.9, 12.86% to 0.7, and a significant 27.76% to 0.5 across four major datasets and two data distributions. The performance decline is more pronounced at a 0.5 pruning ratio, indicating substantial compromises in performance for halving the model parameters.

In the right part of Figure 4, we evaluate the trade-off between model size and accuracy. Assuming the total global model parameters as $N$ and accuracy as Acc, the global pruning ratio as $r$, we weigh the local model parameters against accuracy using a factor $\alpha := N/\text{Acc} > 0$, where the x-axis represents $\text{Acc} + \alpha/r$. A higher $\alpha$ indicates a focus on reducing parameter numbers for large global models, accepting some performance loss. This becomes increasingly advantageous with higher $\alpha$ values, suggesting a promising area for future exploration, especially with larger-scale models.

### 4.3.2 EXPLORING CLIENT-WISE LOCAL PRUNING STRATEGIES

Next, we are interested in exploring the influence of different local pruning strategies. Building upon our initial analysis, we investigate scenarios where our framework permits varying levels of local network pruning ratios. Noteworthy implementations in this domain resemble FjORD (Horváth

Table 2: Comparison of different network local pruning strategies. Global pruning ratio $p$ is 0.9.

| Strategies | CIFAR10 | CIFAR100 | EMNIST-L | FashionMNIST |
|---|---|---|---|---|
| Fixed | 67.65 / 61.17 | 65.41 / 57.38 | 88.75 / 86.33 | 81.75 / 84.27 |
| Uniform ($p = 0.9$) | 65.51 / 60.10 | 64.33 / 58.20 | 85.14 / 84.29 | 78.81 / 77.24 |
| Ordered Dropout ($p = 0.9$) | 61.73 / 58.82 | 61.11 / 53.28 | 82.54 / 80.18 | 75.45 / 73.27 |
| Uniform ($p = 0.7$) | 60.78 / 56.41 | 60.35 / 54.88 | 77.39 / 75.82 | 72.66 / 70.37 |
| Ordered Dropout ($p = 0.7$) | 58.90 / 53.38 | 59.72 / 50.03 | 72.19 / 70.30 | 70.21 / 67.58 |

et al., 2021), FedRolex (Alam et al., 2022), and Flado (Liao et al., 2023). Given that the only partially open-source code available is from FjORD, we employ their layer-wise approach to network sparsity. The subsequent comparisons and their outcomes are presented in Table2. The details of different pruning strategies, including `Fixed`, `Uniform` and `Ordered Dropout` are presented in the above Approach section. "Fixed", "Uniform", "Ordered Dropout" represents *Fixed without pruning*, *Uniform pruning*, and *Uniform order dropout* in the Approach section, respectively. From the results in Table. 2, we can see the difference between `Uniform` and `Ordered Dropout` strategies will be smaller with small global pruning ratio $p$ from 0.9 to 0.7. Besides, in our experiments, `Ordered Dropout` is no better than the simple `Uniform` strategy for local pruning.

### 4.3.3 EXPLORING ADAPTIVE MODEL AGGREGATION STRATEGIES

In this section, we explore a range of weighting strategies, including both simple and advanced averaging methods, primarily focusing on the CIFAR10/100 datasets. We assign clients with $1 - 3$ layers (`OPU1-2-3`) or $2 - 3$ layers (`OPU2-3`) randomly. In Algorithm 3, we implement two aggregation approaches: `simple` and `weighted` aggregation.

Let $L^l$ denote the set of clients involved in training the $l$-th layer, where $l \in \mathcal{L}$. The server's received weights for layer $l$ from client $i$ are represented as $W_{t,K,i}^l$. The general form of model aggregation is thus defined as:

$$W_{t+1}^l = \sum_{j=1}^{L^l} \alpha_i W_{t,K,i}^l.$$

If $\alpha_i$ is initialized as $1/|L^l|$, this constitutes `simple` mean averaging. Considering $N_i$ as the total number of layers for client $i$ and $n$ as the total number of clients, if $\alpha_i = N_i / \sum_{j=1}^{n} N_j$, this method is termed `weighted` averaging. The underlying idea is that clients with more comprehensive network information should have greater weight in parameter contribution. A more flexible approach is `attention` averaging, where $\alpha_i$ is learnable, encompassing `simple` and `weighted` averaging as specific cases.

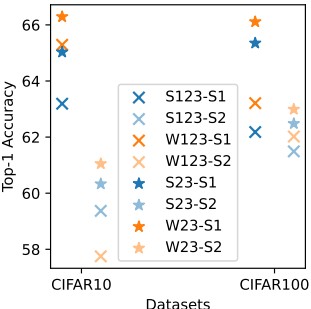

Figure 5: Comparison of various model aggregation strategies. $p = 0.9$.

Future research may delve into a broader range of aggregation strategies. Our findings, shown in Figure 5, include `S123-S1` for the `OPU1-2-3` method with simple aggregation in class-wise non-iid distributions, and `W23-S2` for `OPU2-3` with weighted aggregation in Dirichlet non-iid. The data illustrates that `weighted` averaging relatively improves over `simple` averaging by 1.01% on CIFAR10 and 1.05% on CIFAR100. Furthermore, `OPU-2-3` consistently surpasses `OPU1-2-3` by 1.89%, empirically validating our hypotheses.

## 5 CONCLUSION

In this paper, we introduce FedP3, a nuanced solution designed to tackle both data and model heterogeneities while prioritizing privacy. We have precisely defined the concepts of personalization, privacy, and pruning as central to our analysis. The efficacy of each component is rigorously validated through comprehensive proofs and extensive experimental evaluations.

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

CONTENTS

# A    EXTENDED RELATED WORK

## A.1    FEDERATED NETWORK PRUNING

We introduce two distinct types of network pruning within our study: 1) global pruning, which extends from server to client, and 2) local pruning, where each client's network is pruned based on its own specific data. In our setting, we assume federated pruning is the scenario with both possible global and local pruning. Federated network pruning, a closely related field, pursues the objective of identifying the optimal or near-optimal pruned neural network at each communication from the server to the clients, as documented in works of Jiang et al. (2022a) and Huang et al. (2022), for example.

During the initial phase of global pruning, (Jiang et al., 2022a) isolates a single potent and reliable client to initiate model pruning. The subsequent stage of local pruning incorporates all clients, advancing the adaptive pruning process. This process involves not only parameter removal but also the reintroduction of parameters, complemented by the standard FedAvg (McMahan et al., 2017). However, the need for substantial local memory to record the updated relevance measures of all parameters in the full-scale model poses a challenge. As a solution to this problem, Huang et al. (2022) proposes an adaptive batch normalization and progressive pruning modules that utilize sparse local computation. Yet, these methods overlook explicit considerations for constraints related to client-side computational resources and communication bandwidth.

Our primary attention gravitates towards designing distinct local pruning methods, such as (Horváth et al., 2021), (Alam et al., 2022), and (Liao et al., 2023). Instead of learning the optimal or suboptimal pruned local network, each client attempts to identify the optimal adaptive sparsity method. The work of Horváth et al. (2021) has been groundbreaking, as they introduced Ordered Dropout to navigate this issue, achieving commendable results. It's noteworthy that our overarching framework is compatible with these methods, facilitating straightforward integration of diverse local pruning methods. There are other noticeable methods, such as (Diao et al., 2021), which focuses on reducing the size of each layer in neural networks. In contrast, our approach contemplates a more comprehensive layer-wise selection and emphasizes neuron-oriented sparsity.

As of our current knowledge, no existing literature directly aligns with our approach, despite its practicality and generality. Even the standard literature regarding federated network pruning appears to be rather constrained.

## A.2    SUBNETWORK TRAINING

Our research aligns with the rising interest in Independent Subnetwork Training (IST), a technique that partitions a neural network into smaller components. Each component is trained in a distributed parallel manner, and the results are subsequently aggregated to update the weights of the entire model. The decoupling in IST enables each subnetwork to operate autonomously, using fewer parameters than the complete model. This not only diminishes the computational cost on individual nodes but also expedites synchronization.

This approach was introduced by Yuan et al. (2022) for networks with fully connected layers and was later extended to ResNets Dun et al. (2022) and Graph architectures Wolfe et al. (2023). Empirical evaluations have consistently posited IST as an attractive strategy, proficiently melding data and model parallelism to train expansive models even with restricted computational resources.

Further theoretical insights into IST for overparameterized single hidden layer neural networks with ReLU activations were presented by Liao & Kyrillidis (2022). Concurrently, Shulgin & Richtárik (2023) revisited IST, exploring it through the lens of sketch-type compression.

While acknowledging the adaptation of IST to FL using asynchronous distributed dropout techniques Dun et al. (2023), our approach diverges significantly from prior works. We advocate that clients should not relay the entirety of their subnetworks to the central server—both to curb excessive networking costs and to safeguard privacy. Moreover, our model envisions each client akin to an assembly line component: each specializes in a fraction of the complete neural network, guided by its intrinsic resources and computational prowess.

In Section A.1 and A.2, we compared our study with pivotal existing research, focusing on federated network pruning and subnetwork training. Responding to reviewer feedback, we have broadened the scope of our related work section to include a more extensive comparison with other significant studies.

## A.3 MODEL HETEROGENEITY

Model heterogeneity denotes the variation in local models trained across diverse clients, as highlighted in previous research (Kairouz et al., 2021; Ye et al., 2023). A seminal work by Smith et al. (2017) extended the well-known COCOA method (Jaggi et al., 2014; Ma et al., 2015), incorporating system heterogeneity by randomly selecting the number of local iterations or mini-batch sizes. However, this approach did not account for variations in client-specific model architectures or sizes. Knowledge distillation has emerged as a prominent strategy for addressing model heterogeneity in Federated Learning (FL). Li & Wang (2019) demonstrated training local models with distinct architectures through knowledge distillation, but their method assumes access to a large public dataset for each client, a premise not typically found in current FL scenarios. Additionally, their approach, which shares model outputs, contrasts with our method of sharing pruned local models. Building on this concept, Lin et al. (2020) proposed local parameter fusion based on model prototypes, fusing outputs of clients with similar architectures and employing ensemble training on additional unlabeled datasets. Tan et al. (2022) introduced an approach where clients transmit the mean values of embedding vectors for specific classes, enabling the server to aggregate and redistribute global prototypes to minimize the local-global prototype distance. He et al. (2021) developed FedNAS, where clients collaboratively train a global model by searching for optimal architectures, but this requires transmitting both full network weights and additional architecture parameters. Our method diverges from these approaches by transmitting only weights from a subset of neural network layers from client to server.

## B EXPERIMENTAL DETAILS

### B.1 STATISTICS OF DATASETS

We provide the statistics of our adopted datasets in Table. 3.

| Dataset | # data | # train per client | # test per client |
|---|---|---|---|
| EMNIST-L (Cohen et al., 2017) | 48K+8K | 392 | 168 |
| FashionMNIST (Xiao et al., 2017) | 60K+10K | 490 | 210 |
| CIFAR10 (Krizhevsky et al., 2009) | 50K+10K | 420 | 180 |
| CIFAR100 (Krizhevsky et al., 2009) | 50K+10K | 420 | 180 |

Table 3: Dataset statistics, with data uniformly divided among 100 clients by default.

### B.2 DATA DISTRIBUTIONS

We emulated non-iid data distribution among clients using both class-wise and Dirichlet non-iid scenarios.

- Class-wise: we designate fixed classes directly to every client, ensuring uniform data volume per class. As specifics, EMNIST-L, FashionMNIST, and CIFAR10 assign 5 classes per client, while CIFAR100 allocates 15 classes for each client.

- Dirichlet: following an approach similar to FedCR (Zhang et al., 2023), we use a Dirichlet distribution over dataset labels to create a heterogeneous dataset. Each client is assigned a vector (based on the Dirichlet distribution) that corresponds to class preferences, dictating how labels–and consequently images–are selected without repetition. This method continues until every data point is allocated to a client. The Dirichlet factor indicates the level of data non-iidness. With a Dirichlet parameter of 0.5, about 80% of the samples for each client on EMNIST-L, FashionMNIST, and CIFAR10 are concentrated in four classes. For CIFAR100, the parameter is set to 0.3.

### B.3 NETWORK ARCHITECTURES

Our primary experiments utilize four widely recognized datasets, with detailed descriptions provided in the Experiments section. For the CIFAR10/100 and FashionMNIST experiments, we opt for CNNs comprising two convolutional layers and four fully-connected layers as our standard network architecture. In contrast, for the EMNIST-L experiments, we employ a four-layer MLP architecture. The specifics of these architectures are outlined in Table 4. Additionally, the default ResNet18 network architecture is selected for our layer-overlapping experiments.

| Layer Type | Size | # of Params. |
|---|---|---|
| Conv + ReLu | $5 \times 5 \times 64$ | 4,864 / 1,664 |
| Max Pool | $2 \times 2$ | 0 |
| Conv + ReLu | $5 \times 5 \times 64$ | 102,464 |
| Max Pool | $2 \times 2$ | 0 |
| FC + ReLu | $1600 \times 1024$ | 1,638,400 |
| FC + ReLu | $1024 \times 1024$ | 1,048,576 |
| FC + ReLu | $1024 \times 10/100$ | 10,240 / 102,400 |

| Layer Type | Size | # of Params. |
|---|---|---|
| FC + ReLu | $784 \times 1024$ | 802,816 |
| FC + ReLu | $1024 \times 1024$ | 1,048,576 |
| FC + ReLu | $1024 \times 1024$ | 1,048,576 |
| FC | $1024 \times 10$ | 10,240 |

Table 4: The left figure depicts the neural network architecture employed for the CIFAR10/100 and FashionMNIST experiments. Conversely, the right figure illustrates the default MLP (Multi-Layer Perceptron) architecture used specifically for the EMNIST-L experiments.

### B.4 TRAINING DETAILS

Our experiments were conducted on NVIDIA A100 or V100 GPUs, depending on their availability in our cluster. The framework was implemented in PyTorch 1.4.0 and torchvision 0.5.0 within a Python 3.8 environment. Our initial code, based on FedCR Zhang et al. (2023), was refined to include hyper-parameter fine-tuning. A significant modification was the use of an MLP network with four `FC` layers for EMNIST-L performance evaluation. We standardized the experiments to 500 epochs with a local training batch size of 48. The number of local updates was set at 10 to assess final performance. For the learning rate, we conducted a grid search, exploring a range from $10^{-5}$ to 0.1, with a fivefold increase at each step. In adapting FedCR, we used their default settings and fine-tuned the $\beta$ parameter across values $0.0001, 0.0005, 0.001, 0.005, 0.01$ for all datasets.

### B.5 QUANTITATIVE ANALYSIS OF REDUCED PARAMETERS

We provide a quantitative analysis of parameter reduction across four datasets, as shown in Figure 6. The x-axis represents different global pruning ratios, and the y-axis indicates the number of parameters. For simplicity, we consider a scenario where, aside from the final fully-connected layer, each client trains only one additional layer, akin to the LowerB method used in our earlier experiments. For instance, the label `FC` refers to a condition where only `FC2` and the final layer are fully trained, with other layers being pruned during server-to-client transfer and dropped in server communication.

With a constant global pruning ratio, the left part of the figure shows the total number of parameters in the locally deployed model post server-to-client pruning, while the right part illustrates the communication cost for each scenario. The numbers atop each bar indicate the relative differences between the largest and smallest elements under various conditions. Across all datasets, we note that higher global pruning ratios result in progressively smaller deployed models. For example, at a 0.5 global pruning ratio, the model size for clients training the `Conv1` layer is 57.93% smaller than those training `FC2`. Moreover, there is a significant disparity in communication costs among clients. The ratios of communication costs are 10815 for CIFAR10, 1522.91 for CIFAR100, 13749.46 for FashionMNIST, and 30.23 for EMNIST-L.

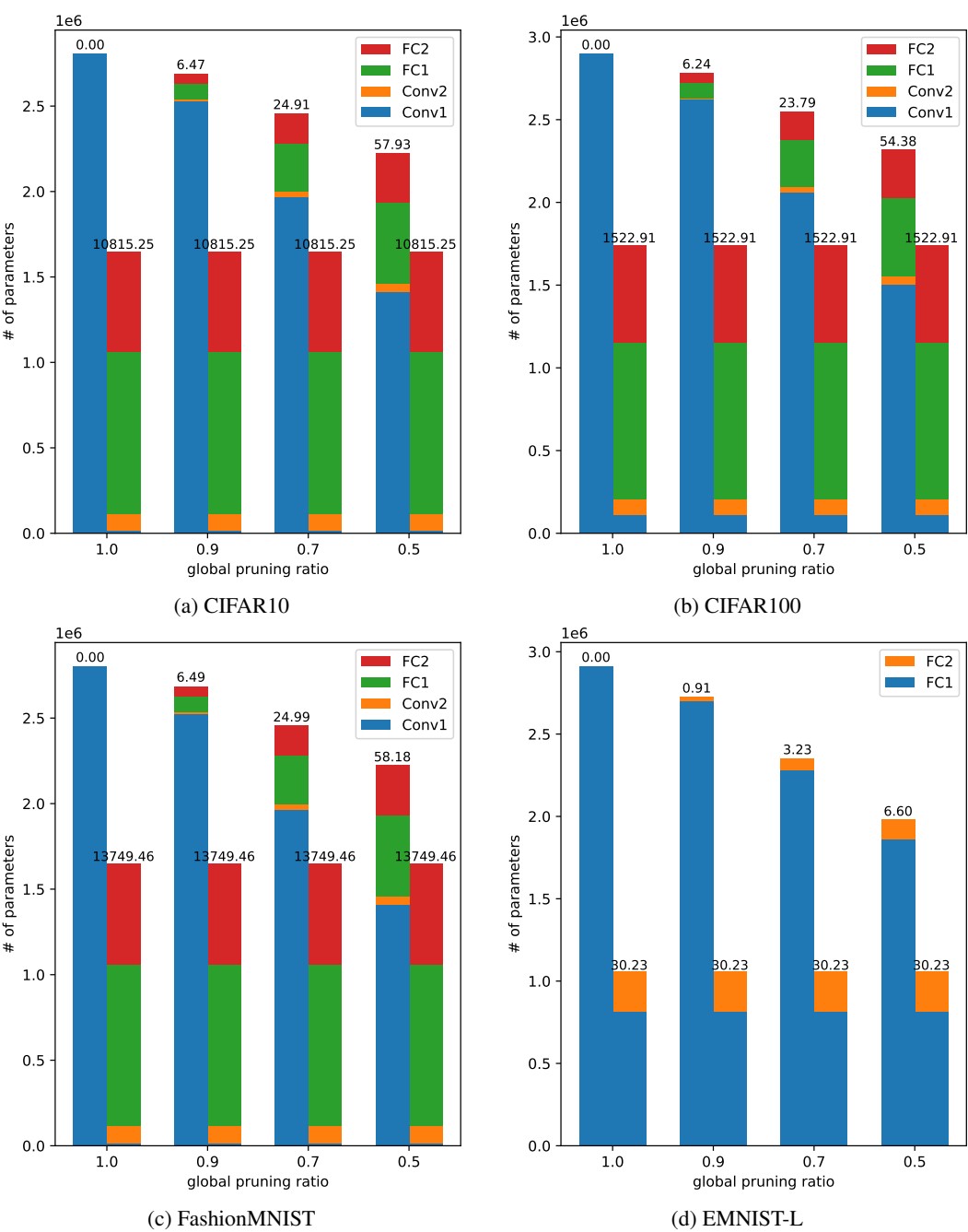

Figure 6: The number of parameters across multiple layers, varying according to different global pruning ratios, spans across four distinct datasets. For each global pruning ratio, the left side of the bar graph shows the total number of parameters in the model after server-to-client pruning when deployed locally. Conversely, the right side details the communication cost associated with each scenario. Atop each bar, we indicate the relative ratio between the layers with the largest and smallest number of parameters, *i.e.,* value $= {}^{(\text{largest}-\text{smallest})}/\text{smallest}$. For (d), since the size of parameters of FC2 and FC3 are the same, we omit plotting FC3 to avoid overlapping.

---

**Algorithm 4** FedP3 theoretical framework

---

1: **Parameters:** learning rate $\gamma > 0$, number of iterations $K$, sequence of global pruning sketches $\left(\mathbf{P}_1^k, \ldots, \mathbf{P}_n^k\right)_{k \leq K}$, aggregation sketches $\left(\mathbf{S}_1^k, \ldots, \mathbf{S}_n^k\right)_{k \leq K}$; initial model $w^0 \in \mathbb{R}^d$
2: **for** $k = 0, 1, \cdots, K$ **do**
3:     Conduct global pruning $\mathbf{P}_i^k w^k$ for $i \in [n]$ and broadcast to all computing nodes
4:     **for** $i = 1, \ldots, n$ in parallel **do**
5:         Compute local (stochastic) gradient w.r.t. personalized model: $\mathbf{P}_i^k \nabla f_i(\mathbf{P}_i^k w^k)$
6:         Take (maybe multiple) gradient descent step $u_i^k = \mathbf{P}_i^k w^k - \gamma \mathbf{P}_i^k \nabla f_i(\mathbf{P}_i^k w^k)$
7:         Send $v_i^k = \mathbf{S}_i^k u_i^k$ to the server
8:     **end for**
9:     Aggregate received subset of layers: $w^{k+1} = \frac{1}{n} \sum_{i=1}^n v_i^k$
10: **end for**

---

## C   EXTENDED THEORETICAL ANALYSIS

### C.1   ANALYSIS OF THE GENERAL FEDP3 THEORETICAL FRAMEWORK

We introduce the theoretical foundation of FedP3, detailed in Algorithm 4. Line 3 demonstrates the global pruning process, employing a biased sketch over randomized sketches $P_i$ for each client $i \in [n]$, as defined in Definition 1. The procedure from Lines 4 to 8 details the local training methods, though we exclude further local pruning for brevity. Notably, our framework could potentially integrate various local pruning techniques, an aspect that merits future exploration.

Our approach uniquely compresses both the weights $w^k$ and their gradients $\nabla f_i(\mathbf{P}_i^k w^k)$. For the sake of clarity, we assume in Line 5 that each client $i$ calculates the pruned full gradient $\mathbf{P}_i^k \nabla f_i(\mathbf{P}_i^k w^k)$, a concept that could be expanded to encompass stochastic gradient computations.

In alignment with Line 6, our subsequent theoretical analysis presumes that each client performs a single-step gradient descent. This assumption stems from observations that local steps have not demonstrated theoretical efficiency gains in heterogeneous environments until very recent studies, such as Mishchenko et al. (2022) and its extensions like Malinovsky et al. (2022); Yi et al. (2023), which required extra control variables not always viable in settings with limited resources.

Diverging from the method in Shulgin & Richtárik (2023), our model involves explicitly sending a selected subset of layers $v_i^k$ from each client $i$ to the server. The aggregation of these layer subsets is meticulously described in Line 9.

Our expanded theoretical analysis is structured as follows: Section C.2 focuses on analyzing the convergence rate of our innovative model aggregation method. In Section C.3, we introduce LDP-FedP3, a novel differential-private variant of FedP3, and discuss its communication complexity in a local differential privacy setting. Section C.4 then delves into the analysis of global pruning, as detailed in Algorithm 4.

### C.2   MODEL AGGREGATION ANALYSIS

In this section, our objective is to examine the potential advantages of model aggregation and to present the convergence analysis of our proposed FedP3. Our subsequent analysis adheres to the standard nonconvex optimization framework, with the goal of identifying an $\epsilon$-stationary point where:

$$\mathbb{E}\left[\|\nabla f(w)\|^2\right] \leq \epsilon, \tag{4}$$

Here, $\mathbb{E}[\cdot]$ represents the expectation over the inherent randomness in $w \in \mathbb{R}^d$. Moving forward, our analysis will focus primarily on the convergence rate of our innovative model aggregation strategy. To begin, we establish the smoothness assumption for each local client's model.

**Assumption 1** (Smoothness)**.** *There exists some $L_i \geq 0$, such that for all $i \in [n]$, the function $f_i$ is $L_i$-smooth, i.e.,*

$$\|\nabla f_i(x) - \nabla f_i(y)\| \leq L_i \|x - y\|, \qquad \forall x, y \in \mathbb{R}^d.$$

This smoothness assumption is very standard for the convergence analysis (Nesterov, 2003; Ghadimi & Lan, 2013; Mishchenko et al., 2022; Malinovsky et al., 2022; Li & Li, 2022; Yi et al., 2023). The smoothness of function $f$ is $\bar{L} = \frac{1}{n}\sum_{i=1}^{n} L_i$, we denote $L_{\max} := \max_{i \in n} L_i$.

We demonstrate the convergence of our proposed FedP3, with a detailed proof presented in Section D.1. Here, we restate Theorem 1 for clarity:

**Theorem 1** (Personalized Model Aggregation). *Let Assumption 1 holds. Iterations $K$, choose step-size $\gamma \leq \left\{ \frac{1}{L_{\max}}, \frac{1}{\sqrt{\hat{L}L_{\max}K}} \right\}$. Denote $\Delta_0 := f(w^0) - f^{\inf}$. Then for any $K \geq 1$, the iterates $w^k$ of FedP3 in Algorithm 4 satisfy*

$$\min_{0 \leq k \leq K-1} \mathbb{E}\left[ \left\| \nabla f(w^k) \right\|^2 \right] \leq \frac{2(1 + \bar{L}L_{\max}\gamma^2)^K}{\gamma K} \Delta_0. \tag{3}$$

Next, we interpret the results. Utilizing the inequality $1 + w \leq \exp(w)$ and assuming $\gamma \leq \frac{1}{\sqrt{\bar{L}L_{\max}K}}$, we derive the following:

$$(1 + \bar{L}L_{\max}\gamma^2)^K \leq \exp(\bar{L}L_{\max}\gamma^2 K) \leq \exp(1) \leq 3.$$

Incorporating this into the equation from Theorem 1, we ascertain:

$$\min_{0 \leq k \leq K-1} \mathbb{E}\left[ \left\| \nabla f(w^k) \right\|^2 \right] \leq \frac{6}{\gamma K} \Delta_0.$$

To ensure the right-hand side of the above equation is less than $\epsilon$, the condition becomes:

$$\frac{6\Delta_0}{\gamma K} \leq \epsilon \Rightarrow K \geq \frac{6\Delta_0}{\gamma \epsilon}.$$

Given $\gamma \leq \frac{1}{\sqrt{\bar{L}L_{\max}K}}$, it follows that $K \geq \frac{36(\Delta_0)^2}{\bar{L}L_{\max}\epsilon^2} = \mathcal{O}\left(\frac{1}{\epsilon^2}\right)$.

Considering the communication cost per iteration is $n \times v = n \times \frac{d}{n} = d$, the total communication cost is:

$$C_{\text{FedP3}} = \mathcal{O}\left(\frac{d}{\epsilon^2}\right).$$

We compare this performance with an algorithm lacking our specific model aggregation design, namely Distributed Gradient Descent (DGD). When DGD satisfies Assumption 4 with $A = C = 0, B = 1$ as per Theorem 5, the total iteration complexity to achieve an $\epsilon$-stationary point is $\mathcal{O}\left(\frac{1}{\epsilon}\right)$. Given that the communication cost per iteration is $nd$, the total communication cost for DGD is:

$$C_{\text{DGD}} = \mathcal{O}\left(\frac{nd}{\epsilon}\right).$$

We observe that the communication cost of FedP3 is more efficient than DGD by a factor of $\mathcal{O}(n/\epsilon)$. This is particularly advantageous in practical Federated Learning (FL) scenarios, where a large number of clients are distributed, highlighting the suitability of our method for such environments. This efficiency also opens avenues for further exploration in large language models.

Although we have demonstrated provable advantages in communication costs for large client numbers, we anticipate that our method's performance exceeds our current theoretical predictions. This expectation is based on the comparison of FedP3 and DGD under Lemma 1. For DGD, with parameters $A = \bar{L}, B = C = 0$, the iteration complexity aligns with $\mathcal{O}(\frac{1}{\epsilon^2})$, leading to a communication cost of:

$$C'_{\text{DGD}} = \mathcal{O}\left(\frac{nd}{\epsilon^2}\right).$$

This indicates a significant reduction in communication costs by a factor of $n$ without additional requirements. It implies that if we could establish a tighter bound on $\|\nabla f_i(w)\|^2$, beyond the scope of Lemma 1, our theoretical results could be further enhanced.

---

**Algorithm 5** Differential-Private FedP3 (LDP-FedP3)

---

1: **Parameters:** learning rate $\gamma > 0$, number of iterations $K$, sequence of aggregation sketches $\left(\mathbf{S}_1^k, \ldots, \mathbf{S}_n^k\right)_{k \leq K}$, perturbation variance $\sigma^2$, minibatch size $b$
2: **for** $k = 0, 1, 2 \ldots$ **do**
3:     Server broadcasts $w^k$ to all clients
4:     **for** each client $i = 1, \ldots, n$ in parallel **do**
5:         Sample a random minibatch $\mathcal{I}_b$ with size $b$ from lcoal dataset $D_i$
6:         Compute local stochastic gradient $g_i^k = \frac{1}{b} \sum_{j \in \mathcal{I}_b} \nabla f_{i,j}(w^k)$
7:         Take (maybe multiple) gradient descent step $u_i^k = w^k - \gamma g_i^k$
8:         Gaussian perturbation to achieve LDP: $\tilde{u}_i^k = u_i^k + \zeta_i^k$, where $\zeta_i^k \sim \mathcal{N}(\mathbf{0}, \sigma^2 \mathbf{I})$
9:         Send $v_i^k = \mathbf{S}_i^k \tilde{u}_i^k$ to the server
10:     **end for**
11:     Server aggregates received subset of layers: $w^{k+1} = \frac{1}{n} \sum_{i=1}^n v_i^k$
12: **end for**

---

## C.3   DIFFERENTIAL-PRIVATE FEDP3 ANALYSIS

The integration of gradient pruning as a privacy preservation method was first brought to prominence by Zhu et al. (2019). Further studies, such as Huang et al. (2020), have delved into the effectiveness of DNN pruning in protecting privacy.

In our setting, we ensure that our training process focuses on extracting partial features without relying on all layers to memorize local training data. This is achieved by transmitting only a select subset of layers from the client to the server in each iteration. By transmitting fewer layers—effectively implementing greater pruning from clients to the server—we enhance the privacy-friendliness of our framework.

This section aims to provide a theoretical exploration of the "privacy-friendly" aspect of our work. Specifically, we introduce a differential-private version of our method, LDP-FedP3, and discuss its privacy guarantees, utility, and communication cost, supported by substantial evidence and rigorous proof.

Local differential privacy is crucial in our context. We aim not only to train machine learning models with reduced communication bits but also to preserve each client's local privacy, an essential element in FL applications. Following the principles of local differential privacy (LDP) as outlined in works like Andrés et al. (2013); Chatzikokolakis et al. (2013); Zhao et al. (2020); Li et al. (2022), we define two datasets $D$ and $D'$ as neighbors if they differ by just one entry. We provide the following definition for LDP:

**Definition 3.** *A randomized algorithm $\mathcal{A} : \mathcal{D} \to \mathcal{F}$, where $\mathcal{D}$ is the dataset domain and $\mathcal{F}$ the domain of possible outcomes, is $(\epsilon, \delta)$-locally differentially private for client $i$ if, for all neighboring datasets $D_i, D_i' \in \mathcal{D}$ on client $i$ and for all events $\mathcal{S} \in \mathcal{F}$ within the range of $\mathcal{A}$, it holds that:*

$$\Pr{\mathcal{A}(D_i) \in \mathcal{S}} \leq e^{\epsilon} \Pr{\mathcal{A}(D_i') \in \mathcal{S}} + \delta.$$

This LDP definition (Definition 3) closely resembles the original concept of $(\epsilon, \delta)$-DP (Dwork et al., 2014; 2006), but in the FL context, it emphasizes each client's responsibility to safeguard its privacy. This is done by locally encoding and processing sensitive data, followed by transmitting the encoded information to the server, without any coordination or information sharing among clients.

Similar to our previous analysis of FedP3, we base our discussion here on the smoothness assumption outlined in Assumption 1. For simplicity, and because our primary focus in this section is on privacy concerns, we assume uniform smoothness across all clients, i.e., $L_i \equiv L$.

Our analysis also relies on the bounded gradient assumption, which is a common consideration in differential privacy analyses:

**Assumption 2** (Bounded gradient). *There exists some constant $C \geq 0$, such that for all clients $i \in [n]$ and for any $x \in \mathbb{R}^d$, the gradient norm satisfies $\|\nabla f_i(x)\| \leq C$.*

Table 5: Comparison of communication complexity in LDP Algorithms for nonconvex problems across distributed settings with $n$ nodes.

| Algorithm | Privacy | Communication Complexity |
|:---:|:---:|:---:|
| Q-DPSGD-1 (Ding et al., 2021) | $(\epsilon, \delta)$-LDP | $\frac{(1+n/(m\tilde{\sigma}^2))m^2\epsilon^2}{d\log(1/\delta)}$ |
| LDP SVRG/SPIDER (Lowy et al., 2023) | $(\epsilon, \delta)$-LDP | $\frac{n^{3/2}m\epsilon\sqrt{d}}{\sqrt{\log(1/\delta)}}$ |
| SDM-DSGD (Zhang et al., 2020) | $(\epsilon, \delta)$-LDP | $\frac{n^{7/2}m\epsilon\sqrt{d}}{(1+\omega)^{3/2}\sqrt{\log(1/\delta)}} + \frac{nm^2\epsilon^2}{(1+\omega)\log(1/\delta)}$ |
| CDP-SGD (Li et al., 2022) | $(\epsilon, \delta)$-LDP | $\frac{n^{3/2}m\epsilon\sqrt{d}}{(1+\omega)^{3/2}\sqrt{\log(1/\delta)}} + \frac{nm^2\epsilon^2}{(1+\omega)\log(1/\delta)}$ |
| LDP-FedP3 (Ours) | $(\epsilon, \delta)$-LDP | $\frac{m\epsilon\sqrt{d}}{\sqrt{\log(1/\delta)}} + \frac{m^2\epsilon^2}{\log(1/\delta)}$ |

This bounded gradient assumption aligns with standard practices in differential privacy analysis, as evidenced in works such as (Bassily et al., 2014; Wang et al., 2017; Iyengar et al., 2019; Feldman et al., 2020; Li et al., 2022).

We introduce a locally differentially private version of FedP3, termed LDP-FedP3, with detailed algorithmic steps provided in Algorithm 5. This variant differs from FedP3 in Algorithm 4 primarily by incorporating the Gaussian mechanism, as per Abadi et al. (2016), to ensure local differential privacy (as implemented in Line 8 of Algorithm 5). Another distinction is the allowance for minibatch sampling per client in LDP-FedP3. Given that our primary focus in this section is on privacy, we set aside the global pruning aspect for now, considering it orthogonal to our current analysis and not central on our privacy considerations. In Theorem 2, we encapsulate the following theorem:

**Theorem 2** (LDP-FedP3). *Under Assumptions 1 and 2, with the use of Algorithm 5, consider the number of samples per client to be $m$ and the number of steps to be $K$. Let the local sampling probability be $q \equiv b/m$. For constants $c'$ and $c$, and for any $\epsilon < c'q^2K$ and $\delta \in (0, 1)$,* LDP-FedP3 *achieves $(\epsilon, \delta)$-LDP with $\sigma^2 = \frac{cKC^2\log(1/\epsilon)}{m^2\epsilon^2}$.*

*Set $K = \max\left\{\frac{m\epsilon\sqrt{L\Delta_0}}{C\sqrt{cd\log(1/\delta)}}, \frac{m^2\epsilon^2}{cd\log(1/\delta)}\right\}$ and $\gamma = \min\left\{\frac{1}{L}, \frac{\sqrt{\Delta_0 cd\log(1/\delta)}}{Cm\epsilon\sqrt{L}}\right\}$, we have:*

$$\frac{1}{K}\sum_{k=0}^{K-1}\mathbb{E}\left[\left\|\nabla f(w^t)\right\|^2\right] \leq \frac{2C\sqrt{Lcd\log(1/\sigma)}}{m\epsilon} = \mathcal{O}\left(\frac{C\sqrt{Ld\log(1/\delta)}}{m\epsilon}\right).$$

*Consequently, the total communication cost is:*

$$C_{\mathrm{LDP-FedP3}} = \mathcal{O}\left(\frac{m\epsilon\sqrt{dL\Delta_0}}{C\sqrt{\log(1/\delta)}} + \frac{m^2\epsilon^2}{\log(1/\delta)}\right).$$

In Section D.2, we provide the proof for our analysis. This section primarily focuses on analyzing and comparing our results with existing literature. Our proof pertains to local differentially-private Stochastic Gradient Descent (SGD). We note that Li et al. (2022) offered a proof for CDP-SGD using a specific set of compressors. However, our chosen compressor does not fall into that category, as discussed more comprehensively in Szlendak et al. (2021). Considering the Rand-$t$ compressor with $t = d/n$, it's established that:

$$\mathbb{E}\left[\left\|\mathcal{R}_t(w) - w\right\|^2\right] \leq \omega\left\|w\right\|^2, \quad \text{where} \quad \omega = \frac{d}{t} - 1 = n - 1.$$

Setting the same $K$ and $\gamma$ and applying Theorem 1 from Li et al. (2022), we obtain:

$$\frac{1}{K}\sum_{k=0}^{K-1}\mathbb{E}\left[\left\|\nabla f(w^t)\right\|^2\right] \leq \frac{5C\sqrt{Lcd\log(1/\sigma)}}{m\epsilon} = \mathcal{O}\left(\frac{C\sqrt{Ld\log(1/\delta)}}{m\epsilon}\right),$$

which aligns with our theoretical analysis. Interestingly, we observe that our bound is tighter by a factor of $2/5$, indicating a more efficient performance in our approach.

We also compare our proposed LDP-FedP3 with other existing algorithms in Algorithm 5. An intriguing finding is that our method's efficiency does not linearly increase with a higher number of clients, denoted as $n$. Notably, our communication complexity remains independent of $n$. This implies that in practical scenarios with a large $n$, our communication costs will not escalate. We then focus on methods with a similar structure, namely, SDM-DSGD and CDP-SGD. For these, the communication cost comprises two components. Considering a specific case, Rand-t, where $t$ is deliberately set to $d/n$, we derive $\omega = d/t - 1 = n - 1$. This results in a communication complexity on par with CDP-SGD, but significantly more efficient than SDM-DSGD. Moreover, it's important to note that the compressor in LDP-FedP3 differs from that in CDP-SGD. Our analysis introduces new perspectives and achieves comparable communication complexity to other well-established results.

### C.4 GLOBAL PRUNING ANALYSIS

Our methodology relates to independent subnetwork training (IST) but introduces distinctive features such as personalization and explicit layer-level sampling for aggregation. IST, although conceptually simple, remains underexplored with only limited studies like Liao & Kyrillidis (2022), which provides theoretical insights for overparameterized single hidden layer neural networks with ReLU activations, and Shulgin & Richtárik (2023), which revisits IST from the perspective of sketch-type compression. In this section, we delve into the nuances of global pruning as applied in Algorithm 4.

For our analysis here, centered on global pruning, we simplify by assuming that all personalized model aggregation sketches $\mathbf{S}_i$ are identical matrices, that is, $\mathbf{S}_i = \mathbf{I}$. This simplification, however, does not trivialize the analysis as the pruning of both gradients and weights complicates the convergence analysis. Additionally, we adhere to the design of the global pruning sketch $\mathbf{P}$ as per Definition 1, which results in a biased estimation, i.e., $\mathbb{E}[\mathbf{P}_i w] \neq w$. Unbiased estimators, such as Rand-t that operates over coordinates, are more commonly studied and offer several advantages in theoretical analysis.

For Rand-t, consider a random subset $\mathcal{S}$ of $[d]$ representing a proper sampling with probability $c_j := \text{Prob}(j \in \mathcal{S}) > 0$ for every $j \in [d]$. $\mathcal{R}_t := \text{Diag}(r_s^1, r_s^2, \cdots, r_s^d)$, where $r_s^j = 1/c_j$ if $j \in \mathcal{S}$ and 0 otherwise. In contrast to our case, the value on each selected coordinate in Rand-t is scaled by the probability $p_i$, equivalent to $|\mathcal{S}|/d$. However, the implications of using a biased estimator like ours are not as well understood.

Our theoretical focus is on Federated Learning (FL) in the context of empirical risk minimization, formulated in (1) within quadratic problem frameworks. This setting involves symmetric matrices $\mathbf{L}_i$, as defined in the following equation:

$$f(w) = \frac{1}{n} \sum_{i=1}^{n} f_i(w), \quad \text{where} \quad f_i(w) \equiv \frac{1}{2} w^\top \mathbf{L}_i w - w^\top b_i. \tag{5}$$

While Equation 5 simplifies the loss function, the quadratic problem paradigm is extensively used in neural network analysis (Zhang et al., 2019; Zhu et al., 2022; Shulgin & Richtárik, 2023). Its inherent complexity provides valuable insights into complex optimization algorithms (Arjevani et al., 2020; Cunha et al., 2022; Goujaud et al., 2022), thereby serving as a robust model for both theoretical examination and practical applications. In this framework, $f(x)$ is $\overline{\mathbf{L}}$-smooth, and $\nabla f(x) = \overline{\mathbf{L}} x - \overline{\mathrm{b}}$, where $\overline{\mathbf{L}} = \frac{1}{n} \sum_{i=1}^{n} \mathbf{L}_i$, and $\overline{\mathrm{b}} := \frac{1}{n} \sum_{i=1}^{n} b_i$.

At this juncture, we introduce a fundamental assumption commonly applied in the theoretical analysis of coordinate descent-type methods.

**Assumption 3** (Matrix Smoothness). *Consider a differentiable function $f : \mathbb{R}^d \to \mathbb{R}$. We say that $f$ is $\mathbf{L}$-smooth if there exists a positive semi-definite matrix $\mathbf{L} \in \mathbb{R}^{d \times d}$ satisfying the following condition for all $x, h \in \mathbb{R}^d$:*

$$f(x + h) \leq f(x) + \langle \nabla f(x), h \rangle + \frac{1}{2} \langle \mathbf{L} h, h \rangle. \tag{6}$$

The classical $L$-smoothness condition, where $\mathbf{L} = L \cdot \mathbf{I}$, is a particular case of Equation equation 6. The concept of matrix smoothness has been pivotal in the development of gradient sparsification methods, particularly in scenarios optimizing under communication constraints, as shown in Safaryan et al. (2021); Wang et al. (2022). We then present our main theory under the interpolation regime for a quadratic problem (5) with $b_i \equiv 0$, as detailed in Theorem 3.

We first provide the theoretical analysis of biased global pruning as implemented in Algorithm 5. To the best of our knowledge, biased gradient estimators have rarely been explored in theoretical analysis. However, our approach of intrinsic submodel training or global pruning is inherently biased. Shulgin & Richtárik (2023) proposed using the Perm-K (Szlendak et al., 2021) as the global pruning sketch. Unlike their approach, which assumes a pruning connection among clients, our method considers the biased Rand-K compressor over coordinates.

**Theorem 3** (Global pruning). *In the interpolation regime for a quadratic problem (5) with $\overline{\mathbf{L}} \succ 0$ and $b_i \equiv 0$, let $\overline{\mathbf{L}}^k := \frac{1}{n} \sum_{i=1}^{n} \mathbf{P}_i^k \overline{\mathbf{L}} \mathbf{P}_i^k$. Assume that $\overline{\mathbf{W}} := \frac{1}{2}\mathbb{E}[\mathbf{P}^k \overline{\mathbf{L}} \overline{\mathbf{B}}^k + \mathbf{P}^k \overline{\mathbf{B}}^k \overline{\mathbf{L}}] \succeq 0$ and there exists a constant $\theta > 0$ such that $\mathbb{E}[\overline{\mathbf{B}}^k \overline{\mathbf{L}} \overline{\mathbf{B}}^k] \preceq \theta \overline{\mathbf{W}}$. Also, assume $f(\mathbf{P}^k w^k) \leq (1 + \gamma^2 h)f(w^k) - f^{\text{inf}}$ for some $h > 0$. Fixing the number of iterations $K$ and choosing the step size $\gamma \in \min\left\{ \sqrt{\frac{\log 2}{hK}}, \frac{1}{\theta} \right\}$, the iterates satisfy:*

$$\mathbb{E}\left[ \|\nabla f(w^k)\|_{\overline{\mathbf{L}}^{-1} \overline{\mathbf{W}} \overline{\mathbf{L}}^{-1}}^2 \right] \leq \frac{4\Delta_0}{\gamma K},$$

*where $\Delta_0 = f(w^0) - f^{\text{inf}}$.*

By employing the definition of $\gamma$, we demonstrate that the iteration complexity is $\mathcal{O}(1/\epsilon^2)$. Compared with the analysis in Shulgin & Richtárik (2023), we allow personalization and do not constrain the global pruning per client to be dependent on other clients. Global pruning is essentially a biased estimator over the global model weights, a concept not widely understood. Our theorem provides insightful perspectives on the convergence of global pruning.

Our theory could also extend to the general case by applying the rescaling trick from Section 3.2 in Shulgin & Richtárik (2023). This conversion of the biased estimator to an unbiased one leads to a general convergence theory. However, this is impractical for realistic global pruning analysis, as it involves pruning the global model without altering each weight's scale. Given that IST and biased gradient estimators are relatively new in theoretical analysis, we hope our analysis could provide some insights.

## D    MISSING PROOFS

### D.1    PROOF OF THEOREM 1

Building on the smoothness assumption of $L_i$ outlined in Assumption 1, the following lemma is established:

**Lemma 1.** *Given that a function $f_i$ satisfies Assumption 1 for each $i \in [n]$, then for any $w \in \mathbb{R}^d$, it holds that*

$$\|\nabla f_i(w)\|^2 \leq 2L_i(f_i(w) - f^{\text{inf}}). \tag{7}$$

*Proof.* Consider $w' = w - \frac{1}{L_i}\nabla f_i(w)$. By applying the $L_i$-smoothness condition of $f$ as per Assumption 1, we obtain

$$f_i(w') \leq f_i(w) + \langle \nabla f_i(w), w' - w \rangle + \frac{L_i}{2}\|\nabla f_i(w)\|^2.$$

Taking into account that $f^{\text{inf}} \leq f_i(w')$, it follows that

$$f^{\text{inf}} \leq f_i(w')$$

$$\leq f_i(w) - \frac{1}{L_i}\|\nabla f_i(w)\|^2 + \frac{1}{2L_i}\|\nabla f_i(w)\|^2$$

$$= f_i(w) - \frac{1}{2L_i}\|\nabla f_i(w)\|^2.$$

Rearranging the terms yields the claimed result. □

Since in this section, we are primarily interested in exploring the convergence of our novel model aggregation design, we set $\mathbf{P}_i^k \equiv \mathbf{I}$ for all $i \in [n]$ and $k \in [K]$. Our analysis focuses on exploring the characteristics of $\mathbf{S}$, which leads to the following theorem.

By the definition of model aggregation sketches in Definition 2, we have $\frac{1}{n}\sum_{i=1}^n \mathbf{S}_i = \mathbf{I}$. Thus, the next iterate can be represented as

$$
\begin{aligned}
w^{k+1} &= \frac{1}{n}\sum_{i=1}^n \mathbf{S}_i^k(w^k - \gamma\nabla f_i(w^k)) \\
&= \frac{1}{n}\sum_{i=1}^n \mathbf{S}_i^k w^k - \gamma \underbrace{\frac{1}{n}\sum_{i=1}^n \mathbf{S}_i^k \nabla f_i(w^k)}_{g^k} \\
&= w^k - \gamma g^k.
\end{aligned}
\tag{8}
$$

Bounding $g^k$ is a crucial part of our analysis. To align with existing works on non-convex optimization, numerous critical assumptions are considered. Extended reading on this can be found in Khaled & Richtárik (2020). Here, we choose the weakest assumption among all those listed in Khaled & Richtárik (2020).

**Assumption 4** (ABC Assumption). *For the second moment of the stochastic gradient, it holds that*

$$
\mathbb{E}\left[\|\mathbf{g}(w)\|^2\right] \le 2A(f(w) - f^{\inf}) + B\|\nabla f(w)\|^2 + C,
\tag{9}
$$

*for certain constants $A, B, C \ge 0$ and for all $w \in \mathbb{R}^d$.*

Note that in order to accommodate heterogeneous settings, we assume a localized version of Assumption 4. Specifically, each $g_i^k \equiv \mathbf{S}_i^k \nabla f_i(w^k)$ is bounded for some constants $A_i, B_i, C_i \ge 0$ and all $w^k \in \mathbb{R}^d$.

**Lemma 2.** *The $g^k$ defined in Eqn. 8 satisfies Assumption 4 with $A = L_{\max}$, $B = C = 0$.*

*Proof.* The proof is as follows:

$$
\begin{aligned}
\mathbb{E}_k\left[\|g^k\|^2\right] &= \mathbb{E}_k\left[\|\frac{1}{n}\sum_{i=1}^n S_i \nabla f_i(w^k)\|^2\right] \\
&= \frac{1}{n}\sum_{i=1}^n \|\nabla f_i(w^k)\|^2 \\
&\le \frac{1}{n}\sum_{i=1}^n 2L_i(f_i(w^k) - f^{\inf}) \\
&\le 2L_{\max}(f(w^k) - f^{\inf}),
\end{aligned}
\tag{10}
$$

where Equation 10 follows from Lemma 1. □

We also recognize certain characteristics of the unbiasedness and upper bound of model aggregation sketches, as elaborated in Theorem 4.

**Theorem 4** (Unbiasedness and Upper Bound of Model Aggregation Sketches). *For any vector $w \in \mathbb{R}^d$, the model aggregation sketch $\mathbf{S}_i$, for each $i \in [n]$, is unbiased, meaning $\mathbb{E}[\mathbf{S}_i w] = w$. Moreover, for any set of vectors $y_1, y_2, \ldots, y_n \in \mathbb{R}^d$, the following inequality is satisfied:*

$$
\mathbb{E}\left[\left\|\frac{1}{n}\sum_{i=1}^n \mathbf{S}_i y_i\right\|^2\right] \le \frac{1}{n}\sum_{i=1}^n \|y_i\|^2.
$$

*Proof.* Consider a vector $x \in \mathbb{R}^d$, where $x_i$ denotes the $i$-th element of $x$. We first establish the unbiasedness of the model aggregation sketch (Definition 1):

$$\mathbb{E}[\mathbf{S}_i x] = n \sum_{j=q(i-1)+1}^{qi} \mathbb{E}[x_{\pi_j} e_{\pi_j}] = n \left( \sum_{j=q(i-1)+1}^{qi} \frac{1}{d} \sum_{i=1}^{d} x_i e_i \right) = \frac{nq}{d} x = x. \tag{11}$$

Next, we examine the second moment:

$$\mathbb{E}\left[\|\mathbf{S}_i x\|^2\right] = n^2 \sum_{j=q(i-1)+1}^{qi} \frac{1}{d} \sum_{i=1}^{d} \|x_i\|^2 = n^2 \frac{q}{d} \|x\|^2 = n \|x\|^2.$$

For all vectors $y_1, y_2, \ldots, y_n \in \mathbb{R}^d$, the following inequality holds:

$$\mathbb{E}\left[\left\|\frac{1}{n}\sum_{i=1}^{n}\mathbf{S}_i y_i\right\|^2\right] = \frac{1}{n^2}\sum_{i=1}^{n}\mathbb{E}\left[\|\mathbf{S}_i y_i\|\right] + \sum_{i\neq j}\mathbb{E}\left[\langle \mathbf{S}_i y_i, \mathbf{S}_j y_j\rangle\right] = \frac{1}{n^2}\sum_{i=1}^{n}\mathbb{E}\left[\|\mathbf{S}_i y_i\|\right] = \frac{1}{n}\sum_{i=1}^{n}\|y_i\|^2. \tag{12}$$

Integrating Equation 11 with Equation 12, we also deduce:

$$\mathbb{E}\left[\left\|\frac{1}{n}\sum_{i=1}^{n}\mathbf{S}_i y_i - \frac{1}{n}\sum_{i=1}^{n}y_i\right\|^2\right] \leq \frac{1}{n}\sum_{i=1}^{n}\|y_i\|^2 - \left\|\frac{1}{n}\sum_{i=1}^{n}y_i\right\|^2. \tag{13}$$

$\square$

We now proceed to prove the main theorem of model aggregation, as presented in Theorem 1. This theorem is restated below for convenience:

**Theorem 1** (Personalized Model Aggregation). *Let Assumption 1 holds. Iterations $K$, choose step-size $\gamma \leq \left\{ 1/L_{\max}, 1/\sqrt{\hat{L}L_{\max}K} \right\}$. Denote $\Delta_0 := f(w^0) - f^{\inf}$. Then for any $K \geq 1$, the iterates $w^k$ of* FedP3 *in Algorithm 4 satisfy*

$$\min_{0\leq k\leq K-1} \mathbb{E}\left[\|\nabla f(w^k)\|^2\right] \leq \frac{2(1 + \bar{L}L_{\max}\gamma^2)^K}{\gamma K}\Delta_0. \tag{3}$$

Our proof draws inspiration from the analysis in Theorem 2 of Khaled & Richtárik (2020) and is reformulated as follows:

**Theorem 5** (Theorem 2 in Khaled & Richtárik (2020)). *Under the assumptions that Assumption 1 and 4 are satisfied, let us choose a step size $\gamma > 0$ such that $\gamma \leq \frac{1}{\bar{L}B}$. Define $\Delta \equiv f(w^0) - f^{\inf}$. Then, it holds that*

$$\min_{0\leq k\leq K-1} \mathbb{E}\left[\|\nabla f(w^k)\|^2\right] \leq \bar{L}C\gamma + \frac{2(1 + \bar{L}\gamma^2 A)^K}{\gamma K}\Delta.$$

Careful control of the step size is crucial to prevent potential blow-up of the term and to ensure convergence to an $\epsilon$-stationary point. Our theory can be seen as a special case with $A = L_{\max}, B = 0, C = 0$, as established in Lemma 2. Thus, we conclude our proof.

### D.2 PROOF OF THEOREM 2

To establish the convergence of the proposed method, we begin by presenting a crucial lemma which describes the mean and variance of the stochastic gradient. Consider the stochastic gradient $g_i^k = \frac{1}{b}\sum_{j\in\mathcal{I}_b}\nabla f_{i,j}(w^k)$ as outlined in Line 6 of Algorithm 5.

**Lemma 3** (Lemma 9 in Li et al. (2022)). *Given Assumption 2, for any client $i$, the stochastic gradient estimator $g_i^k$ is an unbiased estimator, that is,*

$$\mathbb{E}_k \left[ \frac{1}{b} \sum_{j \in \mathcal{I}_b} \nabla f_{i,j}(w^k) \right] = \nabla f_i(w^k),$$

*where $\mathbb{E}_k$ denotes the expectation conditioned on all history up to round $k$. Letting $q = \frac{b}{m}$, the following inequality holds:*

$$\mathbb{E}_k \left[ \left\| \frac{1}{b} \sum_{j \in \mathcal{I}_b} \nabla f_{i,j}(w^k) - \nabla f_i(w^k) \right\|^2 \right] \leq \frac{(1-q)C^2}{b}.$$

Considering the definition of $\mathcal{S}_i^k$, we observe that $\frac{1}{n} \sum_{i=1}^n \mathcal{S}_i^k = \mathbf{I}$. According to Algorithm 5, the next iteration $w^{k+1}$ of the global model is given by:

$$w^{k+1} = \frac{1}{n} \sum_{i=1}^n \mathcal{S}_i^k \left( w^k - \gamma g_i^k + \zeta_i^k \right) = w^k - \underbrace{\frac{1}{n} \sum_{i=1}^n \mathcal{S}_i^k (\gamma g_i^k - \zeta_i^k)}_{G^k}.$$

Employing the smoothness Assumption 1 and taking expectations, we derive:

$$\mathbb{E}_k[f(w^{k+1})] \leq f(w^k) - \mathbb{E}_k \left\langle \nabla f(w^k), G^k \right\rangle + \frac{L}{2} \mathbb{E}_k \left\| G^k \right\|^2. \tag{14}$$

Given that $\zeta_i^k \sim \mathcal{N}(\mathbf{0}, \sigma^2 \mathbf{I})$, we have $\mathbb{E}_k[\zeta_i^k] = 0$. Consequently, we can analyze $\mathbb{E}_k \langle \nabla f(w^k), G^k \rangle$ as follows:

$$\mathbb{E}_k \langle \nabla f(w^k), G^k \rangle = \mathbb{E}_k \left\langle \nabla f(w^k), \frac{1}{n} \sum_{i=1}^n \mathcal{S}_i^k (\gamma g_i^k - \zeta_i^k) \right\rangle$$

$$\stackrel{(11)}{=} \mathbb{E}_k \left\langle \nabla f(w^k), \frac{1}{n} \sum_{i=1}^n (\gamma g_i^k - \zeta_i^k) \right\rangle$$

$$= \mathbb{E}_k \left\langle \nabla f(w^k), \gamma \frac{1}{n} \sum_{i=1}^n g_i^k \right\rangle$$

$$\stackrel{(3)}{=} \gamma \left\| \nabla f(w^k) \right\|^2. \tag{15}$$

To bound the last term $\mathbb{E}_k \left\| G^k \right\|^2$ in Equation 14, we proceed as follows:

$$\mathbb{E}_k \left\| G^k \right\|^2 = \mathbb{E}_k \left\| \frac{1}{n} \sum_{i=1}^n \mathcal{S}_i^k \underbrace{(\gamma g_i^k - \zeta_i^k)}_{M_i^k} \right\|^2$$

$$\overset{(12)}{\leq} \frac{1}{n} \sum_{i=1}^n \mathbb{E}_k \left\| M_i^k \right\|^2$$

$$= \frac{1}{n} \sum_{i=1}^n \mathbb{E}_k \left\| \gamma g_i^k - \zeta_i^k \right\|^2$$

$$= \frac{1}{n} \sum_{i=1}^n \mathbb{E}_k \left\| \gamma g_i^k \right\|^2 + d\sigma^2$$

$$= \gamma^2 \frac{1}{n} \sum_{i=1}^n \mathbb{E}_k \left\| g_i^k - \nabla f_i(w^k) + \nabla f_i(w^k) \right\|^2 + d\sigma^2$$

$$\leq \frac{1}{n} \sum_{i=1}^n \gamma^2 \left\| \nabla f_i(w^k) \right\|^2 + \gamma^2 \frac{1}{n} \sum_{i=1}^n \mathbb{E}_k \left\| g_i^k - \nabla f_i(w^k) \right\|^2 + d\sigma^2$$

$$\overset{(3,2)}{\leq} \gamma^2 C^2 + \frac{\gamma^2(1-q)C^2}{b} + d\sigma^2. \tag{16}$$

Incorporating Equations 16 and 15 into Equation 14, we obtain the following inequality for the expected function value at the next iteration:

$$\mathbb{E}_k[f(w^{k+1})] \leq f(w^k) - \gamma \left\| \nabla f(w^k) \right\|^2 + \frac{L}{2} \left( \gamma^2 C^2 + \frac{\gamma^2(1-q)C^2}{b} + d\sigma^2 \right). \tag{17}$$

Before proceeding further, it is pertinent to consider the privacy guarantees of FedP3, which are based on the analysis of SoteriaFL as presented in Theorem 2 of Li et al. (2022). We reformulate this theorem as follows:

**Theorem 6** (Theorem 2 in Li et al. (2022)). *Assume each client possesses $m$ data points. Under Assumption 3 in Li et al. (2022) and given two bounding constants $C_A$ and $C_B$ for the decomposed gradient estimator, there exist constants $c$ and $c'$. For any $\epsilon < c' \frac{b^2 T}{m^2}$ and $\delta \in (0,1)$, SoteriaFL satisfies $(\epsilon, \delta)$-Local Differential Privacy (LDP) if we choose*

$$\sigma_p^2 = \frac{c \left( C_A^2/4 + C_B^2 \right) K \log(1/\delta)}{m^2 \epsilon^2}.$$

In the absence of gradient shift consideration within SoteriaFL, the complexity of the gradient estimator can be reduced. We simplify the analysis by substituting the two bounds $C_A$ and $C_B$ with a single constant $C$. Following a similar setting, we derive the privacy guarantee for LDP-FedP3 as:

$$\sigma^2 = \frac{cC^2 K \log(1/\delta)}{m^2 \epsilon^2}, \tag{18}$$

which establishes that LDP-FedP3 is $(\epsilon, \delta)$-LDP compliant under the above condition.

Substituting $\sigma$ from Equation 18 and telescoping over iterations $k = 1, \ldots, K$, we can demonstrate the following convergence bound:

$$\frac{1}{K} \sum_{k=1}^K \mathbb{E} \left[ \left\| \nabla f(w^k) \right\|^2 \right] \leq \frac{f(w^0) - f^\star}{\gamma K} + \frac{L}{2} \left[ \gamma C^2 + \frac{\gamma(1-q)C^2}{b} + \frac{cdC^2 T \log(1/\delta)}{\gamma m^2 \epsilon^2} \right]$$

$$\leq \frac{\Delta_0}{\gamma K} + \frac{L}{2} \left[ \frac{\gamma(b+1-q)}{b} C^2 + \frac{cdC^2 K \log(1/\delta)}{\gamma m^2 \epsilon^2} \right]$$

$$\leq \frac{\Delta_0}{\gamma K} + \frac{L}{2} \left[ \gamma C^2 + \frac{cdC^2 K \log(1/\delta)}{\gamma m^2 \epsilon^2} \right].$$

To harmonize our analysis with existing works, such as `CDP-SGD` proposed by Li et al. (2022), which compresses the gradient and performs aggregation on the server over the gradients instead of directly on the weights, we reframe Algorithm 5 accordingly. The primary modification involves defining $M_i^k := \gamma g_i^k - \gamma \zeta_i^k$, where $\zeta_i^k$ is scaled by a factor of $\gamma$. This leads to the following convergence result:

$$\frac{1}{K} \sum_{k=1}^{K} \mathbb{E}\left[\left\|\nabla f(w^k)\right\|^2\right] \leq \frac{\Delta_0}{\gamma K} + \frac{\gamma L C^2}{2}\left[1 + \frac{cdK \log(1/\delta)}{m^2 \epsilon^2}\right]. \tag{19}$$

Optimal choices for $K$ and $\gamma$ that align with this convergence result can be defined as:

$$\gamma K = \frac{m\epsilon\sqrt{\Delta_0}}{C\sqrt{Lcd\log(1/\delta)}}, \quad K \geq \frac{m^2 \epsilon^2}{cd\log(1/\delta)}. \tag{20}$$

Adhering to the relationship established in Equation equation 20 and considering the stepsize constraint $\gamma \leq \frac{1}{L}$, we define:

$$K = \max\left\{\frac{m\epsilon\sqrt{L\Delta_0}}{C\sqrt{cd\log(1/\delta)}}, \frac{m^2\epsilon^2}{cd\log(1/\delta)}\right\},$$

$$\gamma = \min\left\{\frac{1}{L}, \frac{\sqrt{\Delta_0 cd\log(1/\delta)}}{Cm\epsilon\sqrt{L}}\right\}.$$

Substituting these into Equation 19, we obtain:

$$\begin{aligned}
\frac{1}{K} \sum_{t=1}^{K} \mathbb{E}\left[\left\|\nabla f(x^t)\right\|^2\right] &\leq \frac{\Delta_0}{\gamma K} + \frac{\gamma L C^2}{2}\left[1 + \frac{cdK\log(1/\delta)}{m^2\epsilon^2}\right] \\
&\leq \frac{\Delta_0}{\gamma K} + \frac{\gamma L C^2 cdK \log(1/\delta)}{m^2\epsilon^2} \\
&= \frac{\Delta_0}{\gamma K} + \frac{\gamma K L C^2 cd\log(1/\delta)}{m^2\epsilon^2} \\
&\leq \frac{2C\sqrt{Lcd\log(1/\delta)}}{m\epsilon} \\
&= \mathcal{O}\left(\frac{C\sqrt{Ld\log(1/\delta)}}{m\epsilon}\right).
\end{aligned}$$

Neglecting the constant $c$, the total communication cost for `LDP-FedP3` is computed as:

$$\begin{aligned}
C_{\text{LDP-FedP3}} &= n\frac{d}{n}K = dK \\
&= \max\left\{\frac{m\epsilon\sqrt{dL\Delta_0}}{C\sqrt{\log(1/\delta)}}, \frac{m^2\epsilon^2}{\log(1/\delta)}\right\} \\
&= \mathcal{O}\left(\frac{m\epsilon\sqrt{dL\Delta_0}}{C\sqrt{\log(1/\delta)}} + \frac{m^2\epsilon^2}{\log(1/\delta)}\right).
\end{aligned}$$

### D.3 PROOF OF THEOREM 3

We consider the scenario where $\mathbf{P}_i^k$ acts as a biased random sparsifier, and $\mathbf{S}_i^k \equiv \mathbf{I}$. In this case, the update rule is given by:

$$w^{k+1} = \frac{1}{n} \sum_{i=1}^{n} \left(\mathbf{P}_i^k w^k - \gamma \mathbf{P}_i^k \nabla f_i(\mathbf{P}_i^k w^k)\right).$$

Let $w \in \mathbb{R}^d$ and let $S$ represent the selected number of coordinates from $d$. Then, $\mathbf{P}_i$ is defined as:

$$\mathbf{P}_i = \mathrm{Diag}(c_s^1, c_s^2, \cdots, c_s^d), \quad \text{where} \quad c_s^j = \begin{cases} 1 & \text{if } j \in S, \\ 0 & \text{if } j \notin S. \end{cases}$$

Given that $\mathbf{P}_i \preceq \mathbf{I}$, it follows that $\frac{1}{n}\sum_{i=1}^n \mathbf{P}_i \preceq \mathbf{I}$.

In the context where $\mathbf{P}_i$ is a biased sketch, we introduce Assumption 5:

**Assumption 5.** *For any learning rate $\gamma > 0$, there exists a constant $h > 0$ such that, for any $\mathbf{P} \in \mathbb{R}^{d \times d}$, $w \in \mathbb{R}^d$, we have:*

$$f(\mathbf{P}w) \le (1 + \gamma^2 h)(f(w) - f^{\mathrm{inf}}).$$

Assumption 5 assumes the pruning sketch is bounded. Given that the function value should remain finite, this assumption is reasonable and applicable.

In this section, for simplicity, we focus on the interpolation case where $f_i(x) = \frac{1}{2} w^\top \mathbf{L}_i w$. The extension to scenarios with $b_i \ne 0$ is left for future work. By leveraging the $\overline{\mathbf{L}}$-smoothness of function $f$ and the diagonal nature of $\mathbf{P}_i$, we derive the following:

$$
\begin{aligned}
f(w^{k+1}) &:= f\left( \frac{1}{n} \sum_{i=1}^n (\mathbf{P}_i^k w^k - \gamma \mathbf{P}_i^k \nabla f_i(\mathbf{P}_i^k w^k)) \right) \\
&= f\left( \underbrace{\frac{1}{n} \sum_{i=1}^n \mathbf{P}_i^k}_{\mathbf{P}^k} w^k - \gamma \underbrace{\frac{1}{n} \sum_{i=1}^n \mathbf{P}_i^k \overline{\mathbf{L}}_i \mathbf{P}_i^k}_{\overline{\mathbf{B}}^k} w^k \right) \\
&\le f(\mathbf{P}^k w^k) - \gamma \langle \nabla f(\mathbf{P}^k w^k), \overline{\mathbf{B}}^k w^k \rangle + \frac{\gamma^2}{2} \left\| \overline{\mathbf{B}}^k w^k \right\|_{\overline{\mathbf{L}}}^2 \\
&\overset{(5)}{\le} a f(w^k) - \gamma \langle \overline{\mathbf{L}} \mathbf{P}^k w^k, \overline{\mathbf{B}}^k w^k \rangle + \frac{\gamma^2}{2} \left\| \overline{\mathbf{B}}^k w^k \right\|_{\overline{\mathbf{L}}}^2 \\
&= a f(w^k) - \gamma (w^k)^\top \mathbf{P}^k \overline{\mathbf{L}} \overline{\mathbf{B}}^k w^k + \frac{\gamma^2}{2} (w^k)^\top \overline{\mathbf{B}}^k \overline{\mathbf{L}} \overline{\mathbf{B}}^k w^k
\end{aligned}
\tag{21}
$$

Considering the conditional expectation and its linearity, along with the transformation properties of symmetric matrices, we obtain:

$$w^\top \overline{\mathbf{L}} w = \frac{1}{2} w^\top \left( \overline{\mathbf{L}} + \overline{\mathbf{L}}^\top \right) w.$$

By defining $\overline{\mathbf{W}} := \frac{1}{2} \mathbb{E}\left[ \mathbf{P}^k \overline{\mathbf{L}} \overline{\mathbf{B}}^k + \mathbf{P}^k \overline{\mathbf{B}}^k \overline{\mathbf{L}} \right]$ and setting the stepsize $\gamma$ to be less than or equal to $\frac{1}{\theta}$, we can derive the following:

$$
\begin{aligned}
\mathbb{E}\big[f(w^{k+1})|w^k\big] &\leq af(w^k) - \gamma(w^k)^\top \mathbb{E}\Big[\mathbf{P}^k\,\overline{\mathbf{L}}\,\overline{\mathbf{B}}^k\Big]\,w^k + \frac{\gamma^2}{2}(w^k)^\top \mathbb{E}\Big[\overline{\mathbf{B}}^k\,\overline{\mathbf{L}}\,\overline{\mathbf{B}}^k\Big]\,w^k \\
&= af(w^k) - \gamma(w^k)^\top\,\overline{\mathbf{W}}\,w^k + \frac{\gamma^2}{2}(w^k)^\top \mathbb{E}\Big[\overline{\mathbf{B}}^k\,\overline{\mathbf{L}}\,\overline{\mathbf{B}}^k\Big]\,w^k \\
&= af(w^k) - \gamma(\nabla f(w^k))^\top\,\overline{\mathbf{L}}^{-1}\,\overline{\mathbf{W}}\,\overline{\mathbf{L}}^{-1}\,\nabla f(w^k) + \frac{\gamma^2}{2}(\nabla f(w^k))^\top\,\overline{\mathbf{L}}^{-1}\,\mathbb{E}\Big[\overline{\mathbf{B}}^k\,\overline{\mathbf{L}}\,\overline{\mathbf{B}}^k\Big]\,\overline{\mathbf{L}}^{-1}\,\nabla f(w^k) \\
&\leq af(w^k) - \gamma(\nabla f(w^k))^\top\,\overline{\mathbf{L}}^{-1}\,\overline{\mathbf{W}}\,\overline{\mathbf{L}}^{-1}\,\nabla f(w^k) + \frac{\gamma^2}{2}(\nabla f(w^k))^\top\,\overline{\mathbf{L}}^{-1}\,\theta\,\overline{\mathbf{W}}\,\overline{\mathbf{L}}^{-1}\,\nabla f(w^k) \\
&= af(w^k) - \gamma\left\|\nabla f(w^k)\right\|^2_{\overline{\mathbf{L}}^{-1}\,\overline{\mathbf{W}}\,\overline{\mathbf{L}}^{-1}} + \frac{\theta\gamma^2}{2}\left\|\nabla f(w^k)\right\|^2_{\overline{\mathbf{L}}^{-1}\,\overline{\mathbf{W}}\,\overline{\mathbf{L}}^{-1}} \\
&= af(w^k) - \gamma(1 - \theta\gamma/2)\left\|\nabla f(w^k)\right\|^2_{\overline{\mathbf{L}}^{-1}\,\overline{\mathbf{W}}\,\overline{\mathbf{L}}^{-1}} \\
&\leq af(w^k) - \frac{\gamma}{2}\left\|\nabla f(w^k)\right\|^2_{\overline{\mathbf{L}}^{-1}\,\overline{\mathbf{W}}\,\overline{\mathbf{L}}^{-1}}.
\end{aligned}
\tag{22}
$$

Our subsequent analysis relies on the following useful lemma:

**Lemma 4.** *Consider two sequences $\{X_k\}_{k\geq 0}$ and $\{Y_k\}_{k\geq 0}$ of nonnegative real numbers satisfying, for each $k \geq 0$, the recursion*
$$
X_{k+1} \leq aX_k - Y_k + c,
$$
*where $a > 1$ and $c \geq 0$ are constants. Let $K \geq 1$ be fixed. For each $k = 0, 1, \ldots, K - 1$, define the probabilities*
$$
p_k := \frac{a^{K-(k+1)}}{S_K}, \quad \text{where} \quad S_K := \sum_{k=0}^{K-1} a^{K-(k+1)}.
$$
*Define a random variable $Y$ such that $Y = Y_k$ with probability $p_k$. Then*
$$
\mathbb{E}[Y] \leq \frac{a^K X_0 - X_K}{S_K} + c \leq \frac{a^K}{S_K}X_0 + c.
$$

*Proof.* We start by multiplying the inequality $Y_k \leq aX_k - X_{k+1} + c$ by $a^{K-(k+1)}$ for each $k$, yielding
$$
a^{K-(k+1)}Y_k \leq a^{K-k}X_k - a^{K-(k+1)}X_{k+1} + a^{K-(k+1)}c.
$$

Summing these inequalities for $k = 0, 1, \ldots, K - 1$, we observe that many terms cancel out in a telescopic fashion, leading to
$$
\sum_{k=0}^{K-1} a^{K-(k+1)}Y_k \leq a^K X_0 - X_K + \sum_{k=0}^{K-1} a^{K-(k+1)}c = a^K X_0 - X_K + S_K c.
$$
Dividing both sides of this inequality by $S_K$, we get
$$
\sum_{k=0}^{K-1} p_k Y_k \leq \frac{a^K X_0 - X_K}{S_K} + c,
$$
where the left-hand side represents $\mathbb{E}[Y]$. $\qquad\square$

Building upon Lemma 4 and employing the inequality $1 + x \leq e^x$, which is valid for all $x \geq 0$, along with the fact that $S_K \geq K$, we can further refine the bound:

$$
\frac{a^K}{S_K} \leq \frac{(1 + (a-1))^K}{K} \leq \frac{e^{(a-1)K}}{K}.
\tag{23}
$$

To mitigate the exponential growth observed in Eqn 23, we choose $a = 1 + \gamma^2 h$ for some $h > 0$. Setting the step size as
$$
\gamma \leq \sqrt{\frac{\log 2}{hK}},
$$

ensures that $\gamma^2 hK \leq \log 2$, leading to

$$\frac{a^K}{S_K} \overset{23}{\leq} \frac{e^{(a-1)K}}{K} \leq \frac{e^{\gamma^2 hK}}{K} \leq \frac{2}{K}.$$

Incorporating Lemma 4 into Eqn 22 and assuming a step size $\gamma \leq \sqrt{\frac{\log 2}{hK}}$ for some $h > 0$, we establish the following result:

$$\mathbb{E}\left[\|\nabla f(w^k)\|^2_{\bar{\mathbf{L}}^{-1}\,\overline{\mathbf{W}}\,\bar{\mathbf{L}}^{-1}}\right] \leq \frac{4\Delta_0}{\gamma K}. \tag{24}$$

