# OpenReview forum: "FedP3: Federated Personalized and Privacy-friendly Network Pruning under Model Heterogeneity"
_ICLR.cc/2024/Conference — ICLR 2024 poster_

### Official Review · Reviewer_45Hg · 2023-10-29

**Soundness:** 3 good
**Presentation:** 3 good
**Contribution:** 4 excellent
**Rating:** 8
**Confidence:** 4

**Summary:**

Federated learning has gained attention for its capability to train global models while maintaining local data privacy. This paper delves into the challenge of client-side model heterogeneity, exacerbated by differences in clients' memory, processing, and network capabilities (system heterogeneity). The proposed FedP3 framework can well address these challenges.

**Strengths:**

- Adaptable Design: FedP3 caters to model diversity by allowing personalization based on each client's specific capacities, including computational, memory, and communication constraints.
- Novel Dual-Pruning Strategy: FedP3 integrates a dual-pruning approach. This encompasses both global pruning (server to client) and local pruning executed by individual clients.
- Strong Privacy Commitment: FedP3 prioritizes user privacy. The design ensures that full client data is kept confidential as only the selected layers are transmitted from the client to the server post-local training.

**Weaknesses:**

- Absence of Theoretical Insights: This work seems lacks a strong theoretical foundation or interpretation.
- More Ablations: The current model does not explore all potential ablations, for instance, different data aggregation techniques.

**Questions:**

Please refer to the weaknesses.

---

> ### Author Response · Authors · 2023-11-17
> **Response to Reviewer 45Hg**
>
> Thank you so much for your valuable feedback! We sincerely hope our response below can address your concerns.
>
> ---
> **W1**: Absence of Theoretical Insights: This work seems lacks a strong theoretical foundation or interpretation.
>
> **A1**: We are currently working on the theoretical analysis and will provide a response before the conclusion of this author-reviewer period.
>
> ---
> **W2**: More Ablations: The current model does not explore all potential ablations, for instance, different data aggregation techniques.
>
> **A2**: We acknowledge this point and have included the results and analysis of various FL aggregation strategies in Section 4.3.3 of our revised paper.

---

### Official Review · Reviewer_8d9P · 2023-10-31

**Soundness:** 2 fair
**Presentation:** 4 excellent
**Contribution:** 2 fair
**Rating:** 3
**Confidence:** 4

**Summary:**

The authors present a "privacy-preserving" pruning mechanism that is tailored for heterogeneous clients/devices. In this approach, only a part of the model is transmitted to the server, saving resources and enhancing privacy.

**Strengths:**

- The paper is well written, presented and easy to understand.
- Tackling device heterogeneity on federated learning is an important problem
- Model pruning and submode research is an important step towards addressing device heterogeneity and our ability to train larger models with federated learning

**Weaknesses:**

The main weaknesses of this paper are:
- The authors claim that this method is "privacy-preserving" and designed to maximise privacy overall. At the same time, there is no evaluation at all wrt to privacy (either analytical or through some empirical attacks). Furthermore, the privacy aspect is not discussed, practically assuming that fewer layers sent -> more privacy. While there might be some correlation there, I would expect these claims to be backed up with some thorough evaluation/analysis.

- Similarly, the authors claim that resources are saved (energy, memory, cpu)  and larger models can be trained. But there is no evaluation wrt to any such savings. There are no numerical results wrt to any energy savings, the memory consumption savings, understanding how wide can the heterogeneity can we have wrt to device capabilities. Finally, It would be great to have a thorough study on the convergence speed.

- Finally, given that there are a number of sub-model FL training methods proposed (the authors cite a few), it would be great if the evaluation could be expanded to compare with the state of the art.

**Questions:**

Please see my comments above.

---

> ### Author Response · Authors · 2023-11-17
> **Response to Reviewer 8d9P (Part 1)**
>
> Thank you for your insightful feedback. We hope our response below can adequately address your concerns.
>
> ---
> **W1**: The authors claim that this method is "privacy-preserving" and designed to maximise privacy overall. At the same time, there is no evaluation at all wrt to privacy (either analytical or through some empirical attacks). Furthermore, the privacy aspect is not discussed, practically assuming that fewer layers sent $\rightarrow$ more privacy. While there might be some correlation there, I would expect these claims to be backed up with some thorough evaluation/analysis.
>
> **A1**: Our "privacy-friendly" method enhances privacy by not sharing the complete network architecture from the client to the server. This strategy effectively hides critical architecture details, reinforcing our privacy stance, which we support analytically. The training process is designed to extract features using only a subset of layers, rather than relying on the entire network that memorizes local data. By transmitting only a limited selection of layers in each iteration, our approach becomes more privacy-friendly.
>
> It is important to note that the concept of gradient pruning as a means of preserving privacy was initially popularized by the foundational work of Zhu et al. in "Deep leakage from gradients" (NeurIPS, 2019). Subsequently, studies such as "Privacy-preserving learning via deep net pruning" by Huang et al. (arXiv:2003.01876, 2020) have further explored the effectiveness of DNN pruning in maintaining privacy. Our framework presents a broad scope for meaningful exploration and lays the groundwork for future research. However, the quantitative assessment of privacy in FL remains a complex and open issue. Exploring the privacy dimensions of our framework is an exciting and valuable direction for forthcoming studies.
>
> ---
> **W2-1**: Similarly, the authors claim that resources are saved (energy, memory, cpu) and larger models can be trained. But there is no evaluation wrt to any such savings. There are no numerical results wrt to any energy savings, the memory consumption savings, understanding how wide can the heterogeneity can we have wrt to device capabilities.
>
> **A2-1**: We specify that clients in our study have a fixed number of layers, not equal parameters from fully-trained layers, mirroring a heterogeneous environment with diverse memory and communication capacities. For example, in CIFAR10, the first convolutional layer contains 4,864 parameters, contrasting sharply with the 1,638,400 parameters in the first fully-connected layer. As demonstrated in Appendix B.4 (Figure 6), clients training FC2+FFC layers, in comparison to those training Conv1+FFC layers, face communication costs that are over 10,815 times higher. Additionally, they encounter 24.91\% and 57.93\% increases in the sizes of deployed models at global pruning ratios of 0.7 and 0.5, respectively. Section 4.3.3 delves into the assignment of varying layer numbers to clients, analyzing different weighting strategies and detailing our experimental approach.
>
> Our primary experiments with CNNs (2 Conv and 4 FC layers) on CIFAR-10/100 and Fashion-MNIST demonstrate network size reduction. In OPU2, where the last FC layer and two randomly chosen layers (2Conv+2FC) are trained without global or local pruning, we find a 40\% reduction in parameter communication compared to the full FedAvg model. OPU3, with 20\% less parameter communication, matches FedAvg's performance. Dual pruning significantly decreases both the deployed (global pruning) and locally trained (local pruning) model sizes. Figure 4 highlights our method's efficiency for larger models in collaborative training, with a detailed quantitative analysis in Figure 6. The revised paper's Section 4.2 also presents an in-depth analysis of ResNet18.
>
> ---
> **W2-2**: Finally, It would be great to have a thorough study on the convergence speed.
>
> **A2-2**: We are currently working on the theoretical analysis and will provide a response before the conclusion of this author-reviewer period.

---

> ### Author Response · Authors · 2023-11-17
> **Response to Reviewer 8d9P (Part 2)**
>
> **W3**: Finally, given that there are a number of sub-model FL training methods proposed (the authors cite a few), it would be great if the evaluation could be expanded to compare with the state of the art.
>
> **A3**: Our framework is not only versatile but also well-suited for integrating existing dual-pruning methods. As an illustration, Table 2 in our paper showcases the effective incorporation of the Ordered Dropout method from FjORD (NeurIPS21) into our system. We have enriched the revised version of our paper with an extended related work section, offering a comprehensive comparison with a wide array of relevant studies, as detailed in Appendix A.
>
> Furthermore, our analysis extends beyond mere comparisons with the standard FedAvg algorithm. We have undertaken a thorough comparative study with the cutting-edge algorithm FedCR. The details of this comparison, highlighting the efficacy and advancements of our framework in relation to state-of-the-art methods, are meticulously presented in Figure 2 of Section 4.2. This set of comparisons underscores the robustness and innovation of our framework in the context of FL.

---

> ### Author Response · Authors · 2023-11-22
> **Gentle Reminder for Response**
>
> Thank you once again for your valuable comments. As the discussion stage is nearing its conclusion, we kindly request your feedback on whether our responses adequately address your concerns. We have included a new set of theoretical analyses pertaining to privacy, convergence, and communication complexity in our general response to all reviewers, titled “New Theoretical Analyses of FedP3 and LDP-FedP3“. More details are also provided in the revised paper. We would greatly appreciate any further feedback you may have.

---

### Official Review · Reviewer_Kwpv · 2023-11-08

**Soundness:** 4 excellent
**Presentation:** 3 good
**Contribution:** 3 good
**Rating:** 6
**Confidence:** 5

**Summary:**

This paper focuses on addressing the model heterogeneity problem in federated learning. Concretely, it proposes an adaptable federated framework that leverages the personalized pruning technique in a privacy-preserving way. Experiments show that FedP3 can deal with data and model heterogeneity and adapt to various pruning strategies.

**Strengths:**

1. An interesting network pruning-based pipeline is proposed. In this pipeline, the size of training parameters can be personalized for each client.

2. Various local pruning and global aggregation strategies are developed, which present high flexibility of the proposed framework.

3. Limitations of this paper including theoretical analysis and LLMs aspects are well discussed.

**Weaknesses:**

1. The relation between privacy-friendly property and network pruning is implicit. It is better to provide a detailed illustration of the reason why existing pruning-based FL methods have privacy concerns. This can help readers to better understand the motivation of the privacy-preserving part.

2. The authors have emphasized the possibility of utilizing the proposed method in LLMs-based scenarios but there is a lack of related experiments supporting that point. Only shallow neural networks and ResNet are considered in the experimental part.

3. To show the improved communication efficiency as mentioned in the 5th paragraph of the Introduction, it is better to provide some quantitive results on the number of communicated parameters.

4. The presentation of the paper should be further enhanced and some parts should be reorganized. For example, the training goal and variable notation should be given in the Preliminary or Methodology section rather than the Introduction section.

5. Lack of empirical comparison with existing FL methods that also address model heterogeneity problem.

6. To solve the model heterogeneity problem, some existing work needs to be discussed, e.g. Knowledge Distillation-based FL [1],  Prototype-based FL [2], and NAS-based FL.

[1] Ensemble Distillation for Robust Model Fusion in Federated Learning, NeurIPS 2020

[2] FedProto: Federated Prototype Learning across Heterogeneous Clients, AAAI 2022

[3] FedNAS: Federated Deep Learning via Neural Architecture Search, arXiv 2020

**Questions:**

1. Please refer to weakness.

2. How to conduct federated training of LLMs based on the proposed method?

3: In the algorithm, predefined pruning mechanisms are assigned for each client. Is this a common operation in FL under model heterogeneity issues? Are there any previous works conducting similar operations?

4: There are multiple pruning and averaging strategies that can be adopted in the proposed FedP3. For a specific case, how to select the most appropriate strategy?

5. In practice, what if we set several types of models to tackle various heterogeneous devices? Can you set a baseline algorithm to test it?

---

> ### Author Response · Authors · 2023-11-17
> **Response to Reviewer Kwpv (Part 1)**
>
> Thanks for your valuable comments. We hope our response below can adequately address your concerns.
>
> ---
> **W1**: The relation between privacy-friendly property and network pruning is implicit. It is better to provide a detailed illustration of the reason why existing pruning-based FL methods have privacy concerns. This can help readers to better understand the motivation of the privacy-preserving part.
>
> **A1**:  Firstly, it is important to note that the concept of gradient pruning as a means of preserving privacy was initially popularized by the foundational work of Zhu et al. in "Deep leakage from gradients" (NeurIPS, 2019). Subsequently, studies such as "Privacy-preserving learning via deep net pruning" by Huang et al. (arXiv:2003.01876, 2020) have further explored the effectiveness of DNN pruning in maintaining privacy.
>
> In our setting, we ensure that our training process focuses on extracting partial features without relying on all layers to memorize local training data. This is achieved by transmitting only a limited subset of layers from the client to the server in each iteration. Under this approach, sending fewer layers—effectively implementing more pruning from clients to the server—enhances the privacy-friendliness of our framework.
>
> ---
> **W2**: The authors have emphasized the possibility of utilizing the proposed method in LLMs-based scenarios but there is a lack of related experiments supporting that point. Only shallow neural networks and ResNet are considered in the experimental part.
>
> **A2**: We appreciate the interest in applying our framework to scenarios involving LLMs, and we plan to address this in future work. The main objective of this paper is to validate the effectiveness of our novel framework and to ascertain whether existing best practices in FL that are orthogonal to our approach can be successfully integrated.
>
> ---
> **W3**: To show the improved communication efficiency as mentioned in the 5th paragraph of the Introduction, it is better to provide some quantitive results on the number of communicated parameters.
>
> **A3**: In our approach, where we randomly select a subset of layers across all clients, we focus on the average communication cost. Our experiments on CIFAR10/100 and FashionMNIST demonstrate significant communication cost savings: 20\% for OPU3, 40\% for OPU2, and 60\% for LowerB. In the case of ResNet18 experiments, the communication reduction is 6.25\% for both -B2(part) and -B3(part), and 12.5\% for -B2-B3(full). We have expanded Section 4.2 to include a more comprehensive analysis. Additionally, Figure 6 illustrates the distribution of communicated parameters across different layers. We observed notable heterogeneity in our model settings. For instance, the relative saving ratio between two clients can exceed 10815 times. Detailed information on this can be found in Appendix B.5.
>
> ---
> **W4**: The presentation of the paper should be further enhanced and some parts should be reorganized. For example, the training goal and variable notation should be given in the Preliminary or Methodology section rather than the Introduction section.
>
> **A4**: In the Introduction, we introduced two optimization objectives: eq (1) illustrates the standard finite-sum, while eq (2) represents our optimization goal in a high-level sense. Our intention is for the comparison of these two objectives to offer intuitive insights to readers. Starting with the standard FL objective and continuing through the related work section, we aim to furnish essential background information for readers less familiar with FL. Meanwhile, our formal and detailed methodologies are comprehensively outlined in Section 3 - Approach. We've provided a revised version and we remain receptive to suggestions for improving the organization of this content based on further discussions.

---

> ### Author Response · Authors · 2023-11-17
> **Response to Reviewer Kwpv (Part 2)**
>
> **W5**: Lack of empirical comparison with existing FL methods that also address model heterogeneity problem.
>
> **A5**: Firstly, it's important to clarify that our main objective is not necessarily to surpass existing methods addressing model heterogeneity. Instead, our focus is on assessing the viability of our novel framework, which integrates personalized and privacy-friendly network pruning in FL. We are encouraged by extensive experiments that validate the effectiveness of our approach; while there may be some compromise in final testing accuracy, our method notably reduces communication costs and supports flexible dual-pruning, enhancing privacy.
>
> Secondly, the concept of dual-pruning, encompassing both global and local pruning, inherently introduces model heterogeneity. Consequently, we view any pruning method as a potential solution for addressing model heterogeneity. In this respect, our framework is orthogonal to existing approaches, allowing for the safe integration of different pruning methods. The details are provided in Sec 4.3.
>
> Lastly, we are also keen on exploring performance enhancements beyond the standard FedAvg. Our ablation studies include the state-of-the-art FL method FedCR (ICML23), with detailed experiments and analyses provided in Figure 2 in the revised version. These studies reveal that our framework remains efficient, achieving promising performance and significantly reducing the number of required parameters.
>
> ---
> **W6**: To solve the model heterogeneity problem, some existing work needs to be discussed, e.g. Knowledge Distillation-based FL [1], Prototype-based FL [2], and NAS-based FL[3].
>
> **A6**: We are grateful to the reviewer for suggesting relevant literature. We have incorporated a comparison of these works in Appendix A, titled "Extended Related Work."
>
> ---
> **Q2**: How to conduct federated training of LLMs based on the proposed method?
>
> **A7**: We have not yet performed experimental verifications on federated LLMs, a field that remains in its early stages and likely warrants dedicated projects. Currently, we must clarify that the application of our method to federated training of LLMs is theoretical. A crucial initial step would involve developing a reliable FedAvg-like method specifically for LLMs. Given that our method, FedP3 as detailed in Algorithm 1, is not limited by the architecture of the global model, it offers the potential for integration into federated LLM frameworks. This integration would aim at enabling personalized and privacy-aware pruning, albeit with necessary adaptive adjustments.
>
> ---
> **Q3**: In the algorithm, predefined pruning mechanisms are assigned for each client. Is this a common operation in FL under model heterogeneity issues? Are there any previous works conducting similar operations?
>
> **A8**: In addressing the question, our focus is twofold: layer-wise assignment for each client and dual-pruning. In our unique scenario, where clients train only a subset of layers, we lack direct references from existing literature. We chose a simple approach where each client trains a fixed set of layers, although our model could, in theory, permit variable layer selection. However, this adjustment may not substantially benefit our contributions. As for dual-pruning, we follow the conventional design of both global and local pruning, maintaining a consistent pruning mechanism without mixing various strategies, in line with standard practices outlined in Section 2.1.
>
> ---
> **Q4**: There are multiple pruning and averaging strategies that can be adopted in the proposed FedP3. For a specific case, how to select the most appropriate strategy?
>
> **A9**: In Figure 5, our analysis demonstrates that weighted averaging consistently outperforms simple averaging, albeit marginally. Table 2 further illustrates that Order Dropout local pruning consistently falls short in Top-1 accuracy when compared to Uniform strategies. This is likely because Uniform pruning encompasses all weights, thereby providing a more comprehensive global perspective. The versatility and adaptability of our framework with different pruning and averaging techniques open up a wide array of opportunities for more targeted experimentation across various datasets and contexts. A key, consistent observation with our FedP3 method is its ability to achieve comparable performance while significantly reducing communication costs and friendly to privacy.

---

> ### Author Response · Authors · 2023-11-17
> **Response to Reviewer Kwpv (Part 3)**
>
> **Q5**: In practice, what if we set several types of models to tackle various heterogeneous devices? Can you set a baseline algorithm to test it?
>
> **A10**: We appreciate the reviewer's interest in training different types of local models. To the best of our knowledge, two possible approaches are knowledge distillation and building model prototypes. However, knowledge distillation operates on loss or other agency, often requiring additional shared data or constraints. Building model prototypes doesn't align with our objective, which is to train a *single* large global model.
>
> When different models with varying semantic meanings of weights are aggregated, as in combining ResNet18 and ResNet50, it presents numerous unresolved challenges. For example, the distinct semantics of each layer in these models render standard mean aggregation impractical. Developing suitable aggregation strategies for such diverse models is complex and could lead to many interesting research projects. While potential new strategies might be adapted to our general framework with some modifications, there is currently a lack of literature on aggregating outputs from local clients with different model types.
>
> In the context of dual-pruning, our FedP3 framework can accommodate various local models. We also acknowledge the importance of mimicking heterogeneous device environments, including varying memory and bandwidth constraints among devices, an approach we have adopted in our research.

---

### Official Review · Reviewer_CVkW · 2023-11-10

**Soundness:** 3 good
**Presentation:** 3 good
**Contribution:** 2 fair
**Rating:** 5
**Confidence:** 3

**Summary:**

The paper focus on the problem of federated learning with model heterogeneity among clients and designs an algorithm that (1) allows each client only to perform training on a small subnetwork and (2) incorporates model pruning between server and clients to meet the different memory and communication constraints of individual clients. For subnetwork selection, the paper allows each client to train a randomly selected subset of layers of the global model. For model pruning, the paper explores two approaches: uniform pruning and uniform-ordered dropout. Finally, for aggregation of clients' updates, the paper explores simple averaging, weighted averaging (with the weight proportional to the number of layers each client trains), and attention averaging.

**Strengths:**

- The problem of designing FL algorithms under model heterogeneity is well-motivated from practice.
- The paper introduces the new ingredient of subnetwork pruning between the server and the clients and investigates the effect of pruning strategies and pruning ratio on the performance of FL.
- The authors performed necessary ablation studies for the proposed algorithm, such as different subnetwork selections, data heterogeneity levels, sizes of total networks, and aggregation methods.

**Weaknesses:**

- There is a discrepancy between the experiment setting and the motivation problem setting in the introduction.
  - The experiment setting does not reflect the heterogeneity of memory and communication constraints between clients, even though this is a major motivation mentioned in the introduction. Specifically, in all experiments, different clients train subnetworks roughly the same size.
  - The subnetwork is not significantly smaller than the global model. For the ResNet-18 experiments, the subnetwork of each client appears to be at least around half the scale of the global model. This differs from the interesting scenario mentioned in the introduction, where each subnetwork is significantly smaller than the global model.
- Lack of baselines for interpreting the significance of the proposed algorithm. One baseline that is not mentioned is an approach that performs model pruning on the global model first and then performs standard FL on the pruned smaller model. This is for understanding the necessity of personalized model pruning among clients.
- Algorithm descriptions sometimes need more clarity; see question 3 for more details.

**Questions:**

1. Could the authors comment on whether the algorithm would perform well if the subnetworks trained by individual clients are significantly smaller than the global model? Table 2 shows that the algorithm performs poorly when each client only trains one layer.
2. Could the authors comment on an alternative approach of global model pruning + standard FL on the pruned model?
3. Several parts of the algorithm could be clarified more.
   - Figure 1 shows that each client still needs to store a large proportion of the unpruned global model. Is that correct?
   - In Algorithm 1, the pruned weights $P_i(W_t^l)$ are not used anywhere later.
   - In eq (2), the objective function $h$ does not appear to be defined for the problem considered in this paper.

---

> ### Author Response · Authors · 2023-11-17
> **Response to Reviewer CVkW (Part 1)**
>
> Thank you for taking the time to review our paper and for your thoughtful comments. Following your suggestions, we have added new results and analysis to the revised version of the paper. We hope that this updated version adequately addresses your concerns.
>
> ---
> **W1**: There is a discrepancy between the experiment setting and the motivation problem setting in the introduction. **a)** The experiment setting does not reflect the heterogeneity of memory and communication constraints between clients, even though this is a major motivation mentioned in the introduction. Specifically, in all experiments, different clients train subnetworks roughly the same size. **b)** The subnetwork is not significantly smaller than the global model. For the ResNet-18 experiments, the subnetwork of each client appears to be at least around half the scale of the global model. This differs from the interesting scenario mentioned in the introduction, where each subnetwork is significantly smaller than the global model.
>
> **A1** Thanks for the insightful questions.
>
> **a**: We specify that clients in our study have a fixed number of layers, not equal parameters from fully-trained layers, mirroring a heterogeneous environment with diverse memory and communication capacities. For example, in CIFAR10, the first convolutional layer contains 4,864 parameters, contrasting sharply with the 1,638,400 parameters in the first fully-connected layer. As demonstrated in Appendix B.4 (Figure 6), clients training FC2+FFC layers, in comparison to those training Conv1+FFC layers, face communication costs that are over 10,815 times higher. Additionally, they encounter 24.91\% and 57.93\% increases in the sizes of deployed models at global pruning ratios of 0.7 and 0.5, respectively.
>
> **b**: Our primary experiments with CNNs (2 Conv and 4 FC layers) on CIFAR-10/100 and Fashion-MNIST demonstrate network size reduction. In OPU2, where the last FC layer and two randomly chosen layers (2Conv+2FC) are trained without global or local pruning, we have a 40\% reduction in parameter communication compared to the full FedAvg model.
> OPU3, with 20\% less parameter communication, matches FedAvg's performance. Dual pruning significantly decreases both the deployed (global pruning) and locally trained (local pruning) model sizes. Figure 4 highlights our method's efficiency for larger models in collaborative training, with a detailed quantitative analysis in Figure 6. The revised paper's Section 4.2 presents an in-depth analysis of ResNet18.
>
> ---
> **W2**: Lack of baselines for interpreting the significance of the proposed algorithm. One baseline that is not mentioned is an approach that performs model pruning on the global model first and then performs standard FL on the pruned smaller model. This is for understanding the necessity of personalized model pruning among clients. **Q2**: Could the authors comment on an alternative approach of global model pruning + standard FL on the pruned model?
>
> **A2**: We actually denote this baseline as to the "Full" method depicted in Table 1 and the "Fixed" method in Table 2. We have updated their captions to clarify this. By default, we set the global pruning ratio to 0.9, which entails transferring approximately 10\% fewer parameters from the server to the client. Both "Full" and "Fixed" represent the implementation of standard FedAvg during local training and subsequent model aggregation. Further details are available in Sections 4.3.1 and 4.3.2 of our revised paper.

---

> ### Author Response · Authors · 2023-11-17
> **Response to Reviewer CVkW (Part 2)**
>
> **W3**: Algorithm descriptions sometimes need more clarity; see question 3 for more details. **Q3**: Several parts of the algorithm could be clarified more. **a)** Figure 1 shows that each client still needs to store a large proportion of the unpruned global model. Is that correct? **b)** In Algorithm 1, the pruned weights are not used anywhere later. **c)** In eq (2), the objective function does not appear to be defined for the problem considered in this paper.
>
> **A3**: Thanks for your valuable questions. Our response is as follows:
>
> **a**: We would like to clarify that this assertion is not correct. Our focus is on retaining mainly the *pruned* global model layers and training a smaller portion of the *unpruned* fully-trained layers. We substantiate this with quantitative analysis in the right panel of Figure 4 and throughout Figure 6.
>
> **b**: Correct. Our primary aim is to showcase the feasibility of training large models with reduced communication costs in a privacy-friendly manner. Consequently, our method does not utilize pruned weights for each client.
>
> **c**: Equation (2) outlines our process in a general form. We have not specified the function $h$ here, as it can refer to various aggregation strategies. More details are provided in Algorithm 3 and Section 4.3.3.
>
> ---
> **Q1**: Could the authors comment on whether the algorithm would perform well if the subnetworks trained by individual clients are significantly smaller than the global model? Table 2 shows that the algorithm performs poorly when each client only trains one layer.
>
> **A4**: We acknowledge some potential misunderstandings. average communication cost reduction of 20\% for OPU3, 40\% for OPU2, and 60\% for LowerB. We provide a detailed analysis in our revised paper. We treat LowerB as the lower bound benchmark for providing analytical insights. OPU3, notably, delivers performance on par with FedAvg, yet requires only 80\% of the parameter communication. Specifically, LowerB exhibits an average 8.53\% drop in performance but benefits from 60\% lower communication costs. The size of the locally deployed and trained model parameters is quantitatively analyzed in the right part of Figure 4 and throughout Figure 6. These figures demonstrate that our method achieves significantly smaller deployed model sizes, effectively balancing accuracy trade-offs, particularly when the global model is large.

---

> ### Comment · Reviewer_CVkW · 2023-11-21
>
> Thanks for the authors' response. My main concerns are addressed.
>
> However, the communication cost gain is still not perfectly clear to me. As other reviewers pointed out, the convergence speed needs to be clarified. Although the clients only train a smaller part of the model, the total communication cost may still be considerable if convergence is slower under the pruning than without pruning. I will keep my score as is for now and will update it based on the authors' upcoming discussion on this issue.

---

> > ### Author Response · Authors · 2023-11-22
> > **Theoretical Analysis of Communication Costs**
> >
> > We are pleased to know that we have resolved your major concerns. In the revised paper, a new set of theoretical analyses has been added, specifically addressing your points of interest.
> >
> > In Theorem 1, we detail the convergence analysis of FedP3 and describe its communication cost as $\mathcal{O}(d/\epsilon^2)$. This approach significantly improves upon the unpruned method by a factor of $\mathcal{O}(n/\epsilon)$ for large $n$ values, making it highly practical and well-suited for scalable FL deployments. Furthermore, by utilizing the same gradient estimation boundary defined in the smoothness assumption (Lemma 1), we have achieved a reduction in communication cost by a factor of $\mathcal{O}(d/n), (d>>n)$. This finding indicates that a more precise upper gradient bound could lead to even greater theoretical improvements in communication efficiency.
> >
> > Our locally differentially private variant, LDP-FedP3, demonstrates considerable improvement over existing methods with comparable complexity structures. It significantly outperforms SDM-DSGD and matches the optimal communication complexity of the standard LDP-DCGD, as reported in [a].
> >
> > Our manuscript meticulously details proofs for each theorem, focusing particularly on the critical aspects of communication complexity. Utilizing the standard convergence criteria, $\mathbb{E}[||\nabla f(w)||^2] \leq \epsilon$, we reveal both the convergence rate and the number of iterations, as elucidated in our proofs. It is crucial to underscore that in compressed gradient descent (CGD) methods, adhering to the above criterion, increased compression invariably leads to more iterations, potentially influencing the convergence rate. Consequently, we consider communication complexity to be the most significant factor in our analysis of CGD-type methods.
> >
> > For a more comprehensive overview, please refer to the section "New Theoretical Analyses of FedP3 and LDP-FedP3" in the general response to all reviewers and the revised paper, which provides a structured guideline and detailed insights.
> >
> > [a] Zhize Li, Haoyu Zhao, Boyue Li, and Yuejie Chi. Soteriafl: A unified framework for private federated learning with communication compression. Advances in Neural Information Processing Systems, 35:4285–4300, 2022

---

### Official Review · Reviewer_mPei · 2023-11-11

**Soundness:** 4 excellent
**Presentation:** 3 good
**Contribution:** 4 excellent
**Rating:** 8
**Confidence:** 4

**Summary:**

The appeal of Federated learning lies in its privacy-aware model training using data held locally on clients. However, the issue of varied client capacities, termed system heterogeneity, complicates its implementation. This paper proposes the FedP3 framework, emphasizing federated, personalized, and privacy-friendly network pruning, to cater to such diverse client scenarios.

**Strengths:**

1.	The proposed system comprehensively considers a holistic management of heterogeneity, i.e., it effectively manages both data and model disparities. It supports data distribution among clients, class-wise or Dirichlet non-iid. Moreover, it accommodates variance in the models between the server-client and among individual clients.
2.	The proposed methods are inspiring for real-world scenarios, including dual pruning that supports both global pruning (from server to client) and local pruning by individual clients; few-layer communication from the clients to the server after local training.
3.	Experiments validate that the proposed FedP3 is both effective and adaptable. It paves the way for personalized and privacy-conscious pruning in a heterogeneous federated setting.

**Weaknesses:**

1.	Few-layer communication can also significantly reduce communication costs and save bandwidth, a detailed numerical analysis would help.
2.	The results on various FL aggregation strategies shall be considered for completeness.
3.	Hyperparameter tuning part is not so crystal. More explanation is needed on how to choose and tune the hyperparameter (maybe via grid-search?) to deliver the best possible results for each model.
4.	Broader Literature Review is expected. While this work focuses primarily on the discussion and analysis of the most relevant and typical works, this approach might overlook other pertinent past research that holds tangential relevance to their study.

**Questions:**

1. How much the few-layer communication can save communication costs and bandwidth?
2. How the other FL aggregation strategies work in combination with FedP3?
3. How to choose and tune the hyperparameter?

---

> ### Author Response · Authors · 2023-11-17
> **Response to Reviewer mPei**
>
> Thank you so much for the positive feedback and valuable comments. We hope the following new analysis and clarifications can address your concerns.
>
> ---
> **W1**: Few-layer communication can also significantly reduce communication costs and save bandwidth, a detailed numerical analysis would help. **Q1**: How much the few-layer communication can save communication costs and bandwidth?
>
> **A1**: In the paragraph "Layer Overlapping Analysis" of Section 4.2 of our revised paper, we assess the impact of few-layer communication on reducing costs. Our findings indicate significant savings: 20\% for OPU3, 40\% for OPU2, and 60\% for LowerB on average. Particularly, OPU3 achieves comparable results to FedAvg with only 80\% of the parameters communicated. This method is also applicable to ResNet18 and EMNIST-L datasets. As detailed in Appendix B.4 (Figure 6), clients training FC2+FFC layers, in comparison to those training Conv1+FFC layers, face communication costs that are over 10,815 times higher. Additionally, they encounter 24.91\% and 57.93\% increases in the sizes of deployed models at global pruning ratios of 0.7 and 0.5, respectively.
>
> ---
> **W2**:The results on various FL aggregation strategies shall be considered for completeness. **Q2**: How the other FL aggregation strategies work in combination with FedP3?
>
> **A2**: Acknowledging this, we've added a detailed examination of various FL aggregation strategies in Section 4.3.3 of our revised paper. We show a slight improvement using weighted averaging based on the relative number of layers on each client.
>
> ---
> **W3**: Hyperparameter tuning part is not so crystal. More explanation is needed on how to choose and tune the hyperparameter (maybe via grid-search?) to deliver the best possible results for each model. **Q3**: How to choose and tune the hyperparameter?
>
> **A3**: We optimized key hyper-parameters, including the local training learning rate and the personalized factor of FedCR, through grid search, while maintaining other parameters at default settings for fair comparison. Comprehensive details of the training procedure can be found in Appendix B.3.
>
> ---
> **W4**: Broader Literature Review is expected. While this work focuses primarily on the discussion and analysis of the most relevant and typical works, this approach might overlook other pertinent past research that holds tangential relevance to their study.
>
> **A4**: In the Related Work section of our paper, we initially compared our study to the most pertinent existing research within our knowledge. Responding to feedback, we have expanded this section to encompass and contrast additional studies, particularly those highlighted by reviewers, along with other relevant papers. Further details are provided in Appendix A.

---

### Official Review · Reviewer_2HRe · 2023-11-12

**Soundness:** 2 fair
**Presentation:** 3 good
**Contribution:** 2 fair
**Rating:** 6
**Confidence:** 4

**Summary:**

This paper highlights the growing interest in federated learning (FL) for its privacy-preserving capabilities. It particularly addresses the challenge of client-side model heterogeneity in FL, driven by variations in client resources. Termed "system heterogeneity," this scenario necessitates customizing a unique model for each client. The paper introduces FedP3, a Federated Personalized and Privacy-friendly Network Pruning Framework, designed to address model heterogeneity effectively. FedP3 can adapt established techniques to specific instances, offering a practical solution.

**Strengths:**

==*== Strengths
+ This work offers an effective and adaptable FL framework FedP3 tailored for model heterogeneity scenarios.
+ The proposed personalized network pruning technique is applicable to diverse scenarios.

**Weaknesses:**

==*== Weaknesses
- The outcomes of the experiment need to be made more convincing.
- Limited in-depth comparison with state-of-the-art solutions.
- Privacy analysis and convergence analysis need to be included.
- Empirical verification of the claimed contributions is necessary.

**Questions:**

Comments:

-	Privacy analysis and convergence analysis need to be included. Indeed, separating model parameters from the network parameter architecture is intuitively beneficial to FL privacy protection, but whether existing gradient reconstruction attacks or other privacy attacks will challenge this personalization technology has not been fully explored. Therefore, it would be better if the authors could analyze the privacy performance empirically or theoretically.

-	Furthermore, the convergence analysis of the proposed method has not given any explanation. For model heterogeneous scenarios, reviewers expect to see rigorous analysis and discussion of the convergence and stability of this method.

-	This paper claims that the proposed personalized pruning technique can well alleviate the system heterogeneity and model heterogeneity problems, but the reviewer has not seen any discussion and numerical results on the system heterogeneity scenario.

-	Limited in-depth comparison with state-of-the-art solutions. In fact, personalized pruning technique is not the first time to be applied to FL to solve efficiency and heterogeneity problems. The following literature needs to be included in the baseline solutions and explain the differences and connections between this paper and them.

[1] Zhou X, Jia Q, Xie R. NestFL: efficient federated learning through progressive model pruning in heterogeneous edge computing[C]//Proceedings of the 28th Annual International Conference on Mobile Computing And Networking. 2022: 817-819.

[2] Li A, Sun J, Li P, et al. Hermes: an efficient federated learning framework for heterogeneous mobile clients[C]//Proceedings of the 27th Annual International Conference on Mobile Computing and Networking. 2021: 420-437.

[3] Pase F, Isik B, Gunduz D, et al. Efficient federated random subnetwork training[C]//Workshop on Federated Learning: Recent Advances and New Challenges (in Conjunction with NeurIPS 2022). 2022.

---

> ### Author Response · Authors · 2023-11-17
> **Response to Reviewer 2HRe (Part 1)**
>
> We thank you for the valuable review and for sharing relevant references. We hope that our following discussion will address your raised weaknesses and comments.
>
> ---
> **W1**: The outcomes of the experiment need to be made more convincing.
>
> **A1**: In response, we have implemented several modifications to enhance the clarity and illustrative quality of our experiments. Details are attached in the general response to all reviewers. Please also find our revised paper with more evidence.
>
> We trust that these enhancements will make our experiments and analyses more transparent and insightful. We are readily available to address any specific queries regarding the experimental components.
>
> ---
> **W2**: Limited in-depth comparison with state-of-the-art solutions.
>
> **A2**: We wish to clarify that the primary objective of our paper is not to surpass existing FL algorithms on popular datasets. Instead, our focus is on introducing a general pruning framework that not only allows for personalization but also operates within a privacy-friendly setting. This claim is substantiated by comprehensive experiments conducted throughout the paper.
>
> Given the general nature of our framework, we are also interested in investigating its applicability to cutting-edge personalized FL methodologies. To this end, we have conducted a comparative analysis with the state-of-the-art FedCR (ICML23). The details of our experimental setup, results, and analysis are thoroughly documented in the right part of Figure 2. Our findings are encouraging, indicating that even when applied to FedCR, our framework maintains comparable performance while significantly reducing the volume of parameters communicated.
>
> ---
> **W3-1**: Privacy analysis ... need to be included. **Q1**: Privacy analysis needs to be included ... whether existing gradient reconstruction attacks or other privacy attacks will challenge this personalization technology has not been fully explored. Therefore, it would be better if the authors could analyze the privacy performance empirically or theoretically.
>
> **A3**: Our "privacy-friendly" method enhances privacy by not sharing the complete network architecture from the client to the server. This strategy effectively hides critical architecture details, reinforcing our privacy stance, which we support analytically. The training process is designed to extract features using only a subset of layers, rather than relying on the entire network that memorizes local data. By transmitting only a limited selection of layers in each iteration, our approach becomes more privacy-friendly.
>
> It is important to note that the concept of gradient pruning as a means of preserving privacy was initially popularized by the foundational work of Zhu et al. in "Deep leakage from gradients" (NeurIPS, 2019). Subsequently, studies such as "Privacy-preserving learning via deep net pruning" by Huang et al. (arXiv:2003.01876, 2020) have further explored the effectiveness of DNN pruning in maintaining privacy.
>
> Our framework presents a broad scope for meaningful exploration and lays the groundwork for future research. However, the quantitative assessment of privacy in FL remains a complex and open issue. Exploring the privacy dimensions of our framework is an exciting and valuable direction for forthcoming studies.
>
> ---
> **W3-2**: ... convergence analysis need to be included. **Q2**: Furthermore, the convergence analysis of the proposed method has not given any explanation. For model heterogeneous scenarios, reviewers expect to see rigorous analysis and discussion of the convergence and stability of this method.
>
> **A4**: We are currently working on the theoretical analysis and will provide a response before the conclusion of this author-reviewer period.

---

> ### Author Response · Authors · 2023-11-17
> **Response to Reviewer 2HRe (Part 2)**
>
> **W4**: Empirical verification of the claimed contributions is necessary.
>
> **A4**: We would like to emphasize all our claimed contributions are clearly stated with enough evidence. Here is the response for all our contributions:
>
> - **Contribution 1: general framework.** "A versatile design that caters to model heterogeneity, enabling tailored personalization based on clients’ local capacities (computational, memory and communication constraints)."
>     - Our general framework, FedP3, detailed in Algorithm 1, uses $L_i$, $P_i$, and $Q_i$ as personalization factors for client $i$, aligning with each client's computational capacity and bandwidth. This framework is central to this paper and the feasibility has been verified through all our experiments.
>
> - **Contribution 2: dual-pruning approach.** "A dual-pruning approach that supports both global pruning (from server to client) and local pruning by individual clients."
>     - We provide empirical evidence of this methodology throughout Sections 4.3.1 and 4.3.2.
>
> - **Contribution 3: Privacy-friendly design.** " A commitment to privacy, ensuring that complete client information is never disclosed to the server as only a few layers are communicated from the clients to the server after local training."
>     - Our approach ensures privacy-friendliness by not sending the full network architecture from client to server. The quantitative efficiency of this method is demonstrated throughout our experimental section.
>
> - **Contribution 4: handling data and model heterogeneity.** "Comprehensive handling of both data and model heterogeneity. We allow data distributed among clients to be class-wise or Dirichlet non-iid. Meanwhile, we allow the server-client models and client-wise models to be different.
>       - Our experiments, detailed in Section 4, focus on non-iid data, including class-wise or Dirichlet cases. After dual-pruning, the local models naturally exhibit heterogeneity, a key focus in our paper.
>
> We also revised the contribution part in the introduction for better clarity.
>
>
> ---
> **Q3**: This paper claims that the proposed personalized pruning technique can well alleviate the system heterogeneity and model heterogeneity problems, but the reviewer has not seen any discussion and numerical results on the system heterogeneity scenario.
>
> **A5**: There seems to be a misunderstanding regarding systems heterogeneity, which, as defined in reference [a] below, pertains to the varying storage, computational, and communication capabilities of each device. Our paper directly addresses this issue throughout. The number of personalized layers $L_i$, alongside the global pruning mechanism $P_i$ and the local pruning mechanism $Q_i$ presented in Algorithm 1, are specifically designed to cater to system heterogeneity. Intuitively, different clients may need to train a vastly different number of parameters.
>
> For example, in CIFAR10, the first convolutional layer contains 4,864 parameters, contrasting sharply with the 1,638,400 parameters in the first fully-connected layer. As demonstrated in Appendix B.4 (Figure 6), clients training FC2+FFC layers, in comparison to those training Conv1+FFC layers, face communication costs that are over 10,815 times higher. Additionally, they encounter 24.91\% and 57.93\% increases in the sizes of deployed models at global pruning ratios of 0.7 and 0.5, respectively. Section 4.3.3 delves into the assignment of varying layer numbers to clients, analyzing different weighting strategies and detailing our experimental approach.
>
> Our primary experiments with CNNs (2 Conv and 4 FC layers) on CIFAR-10/100 and Fashion-MNIST demonstrate network size reduction. In OPU2, where the last FC layer and two randomly chosen layers (2Conv+2FC) are trained without global or local pruning, we find a 40\% reduction in parameter communication compared to the full FedAvg model. OPU3, with 20\% less parameter communication, matches FedAvg's performance. Dual pruning significantly decreases both the deployed (global pruning) and locally trained (local pruning) model sizes. Figure 4 highlights our method's efficiency for larger models in collaborative training, with a detailed quantitative analysis in Figure 6. The revised paper's Section 4.2 also presents an in-depth analysis of ResNet18.
>
> [a] Li, Tian, Anit Kumar Sahu, Ameet Talwalkar, and Virginia Smith. "Federated learning: Challenges, methods, and future directions." IEEE signal processing magazine 37, no. 3 (2020): 50-60.

---

> ### Author Response · Authors · 2023-11-17
> **Response to Reviewer 2HRe (Part 3)**
>
> **Q4**: Limited in-depth comparison with state-of-the-art solutions. In fact, personalized pruning technique is not the first time to be applied to FL to solve efficiency and heterogeneity problems. The following literature needs to be included in the baseline solutions and explain the differences and connections between this paper and them.
>
> **A4**: Firstly, it's important to clarify that our main objective is not necessarily to surpass existing methods addressing model heterogeneity. Instead, our focus is on assessing the viability of our novel framework, which integrates personalized and privacy-friendly network pruning in FL. We are encouraged by extensive experiments that validate the effectiveness of our approach; while there may be some compromise in final testing accuracy, our method notably reduces communication costs and supports flexible dual-pruning, enhancing privacy.
>
> Secondly, the concept of dual-pruning, encompassing both global and local pruning, inherently introduces model heterogeneity. Consequently, we view any pruning method as a potential solution for addressing model heterogeneity. In this respect, our framework is orthogonal to existing approaches, allowing for the safe integration of different pruning methods. The details are provided in Sec 4.3.
>
> Lastly, we are also keen on exploring performance enhancements beyond the standard FedAvg. Our ablation studies include the state-of-the-art FL method FedCR (ICML23), with detailed experiments and analyses provided in Figure 2 in the revised version. These studies reveal that our framework remains efficient, achieving commendable performance while significantly reducing the number of required parameters.

---

> > ### Comment · Reviewer_2HRe · 2023-11-19
> > **Response to Authors**
> >
> > Thanks to the authors for their detailed responses. The authors have addressed most of my concerns. However, the following issues still require further clarification:
> >
> > - I look forward to seeing the authors provide more formal privacy analysis before the end of the rebuttal.
> > - The authors may have some misunderstandings about system heterogeneity. As far as I understand, the authors at least need to verify the proposed algorithm under different devices, different network states, and different communication environments.
> >
> > Since the authors have addressed most of my concerns, I will increase my score appropriately.

---

> > > ### Author Response · Authors · 2023-11-22
> > > **Formal Privacy Analysis and System Heterogeneity**
> > >
> > > We are happy to know that we have addressed most of your concerns. Here, we would like to provide feedback with respect to the remaining concerns regarding formal privacy analysis and system heterogeneity.
> > >
> > >
> > > - **Formal privacy analysis.** In Algorithm 5, we introduced LDP-FedP3, a locally differentially private version of FedP3. Theorem 2 details its privacy guarantees, convergence analysis, and total communication cost. Additionally, we discussed how it compares favorably with existing methods. We trust this comprehensive analysis sufficiently addresses your privacy concerns. Further theoretical results are included in the section "New Theoretical Analyses of FedP3 and LDP-FedP3", as mentioned in our general response to all reviewers.
> > >
> > > - **System Heterogeneity.** Our main objective is to assess the effectiveness of our novel FedP3 method across commonly used datasets within settings that encompass both data and model heterogeneity. This is aimed at offering insights for future FL research. We do not concentrate on the development of engineering-intensive systems for a variety of real-world devices, network conditions, and communication environments. Instead, we simulate different processes as distinct clients, a common and widely accepted approach in FL research. Each simulated client has unique characteristics, including model heterogeneity, data heterogeneity, and personalized settings such as global pruning, local pruning, and layer-wise aggregation strategies. Furthermore, we provide theoretical insights into both our method and its locally differentially private variant. We believe our simulations effectively demonstrate the efficacy of our proposed method in both theoretical and experimental contexts.

---

> ### Comment · Reviewer_2HRe · 2023-11-22
> **Response to Authors**
>
> Thanks to the authors for their responses. It's nice to see the authors provide details of the analysis in response to my concerns, and I will revise my score appropriately.

---

### Author Response · Authors · 2023-11-17
**Rebuttal Summary**

We sincerely thank all reviewers for their valuable comments and suggestions. We have made the following updates:

- We have improved visual representation by transforming Table 2 into Figure 2, and now provide an in-depth analysis in Section 4.2 titled "Layer Overlapping Strategies."
- We incorporated a comparative analysis with the leading personalized FL method, FedCR (ICML23), in both Figure 2 and Section 4.2.
- The section "Larger Network Verifications" has been augmented with additional quantitative analyses.
- The benefits of global pruning have been numerically quantified and are now illustrated in Figure 4. Corresponding updates have been made to the analysis in Section 4.3.1.
- A thorough comparative study of various weighting strategies is now included in Section 4.3.3.
- We have expanded the discussion of related work in Appendix A.
- Information regarding dataset statistics has been relocated to Appendix B.1, while data distribution details are now in Appendix B.2.
- A detailed description of the experimental setup, encompassing network architectures and training procedures, is provided in Appendices B.3 and B.4.
- An extensive quantitative analysis of reduced parameters is now included in Appendix B.5.
- Other minor adjustments have been made throughout the manuscript to enhance overall clarity.

---

### Public Comment · ~Sara_Babakniya1 · 2023-11-21
**Another related work**

Dear Authors,

First, I congratulate you on this exciting paper. Also, I would like to mention our recent related paper, FLASH, where we employ pruning to improve efficiency and cost in federated learning. Our work also supports heterogeneous settings (different model capacities and non-IID data distribution). As presented in the paper, FLASH can achieve SOTA performance in terms of accuracy and bidirectional communication costs in various settings. It would be great to see comparisons of FedP3 with FLASH.

Babakniya, S., Kundu, S., Prakash, S., Niu, Y. and Avestimehr, S. "Revisiting Sparsity Hunting in Federated Learning: Why does Sparsity Consensus Matter?" Transactions on Machine Learning Research (2023). (https://openreview.net/forum?id=iHyhdpsnyi). Initial results were presented in the NeurIPS Workshop on Recent Advances and New Challenges in Federated Learning (FL-NeurIPS), 2022 (https://openreview.net/forum?id=XEQSP1zL2gx).

Best,
Sara

---

> ### Author Response · Authors · 2023-11-22
> **Explaining the Differences Between FLASH and FedP3, Adding FLASH to the Related Work**
>
> We would like to express our gratitude for the opportunity to discuss our work in light of the intriguing contributions made by FLASH. Our objective is to underscore the unique aspects and significant advancements of our research, particularly in the context of FL.
>
> - **Innovative Approach in FL**: Our research pioneers the concept of explicit layer-wise communication in FL. This approach stands apart from standard network pruning techniques. Our method addresses the complex challenge of varying semantic information across different layers, which can potentially disrupt global model learning. We believe that the collaborative nature of FL can make this innovative pruning technique feasible, thereby offering a fresh perspective on reducing communication costs.
>
> - **Orthogonality to Dual-Pruning Methods**: Our work exists in a unique space compared to dual-pruning methods. Notably, FLASH does not incorporate local pruning, an aspect we consider crucial in our approach.
>
> - **Rigorous Theoretical Analysis**: We provide a comprehensive theoretical analysis that covers both global pruning from client to server and the nuances of personalized layer-wise aggregation. Our analysis clearly delineates the advantages of our approach, setting it apart from FLASH, which lacks a similar theoretical foundation.
>
> - **Privacy Considerations with LDP-FedP3**: Our study introduces a local differential-private variant, LDP-FedP3. This variant is accompanied by detailed discussions on privacy guarantees, convergence, and communication costs. We also compare it with existing methods to highlight its benefits. This aspect of privacy concern is not addressed in FLASH.
>
> - **Data Heterogeneity Settings (minor)**: Our research explores two distinct data heterogeneity scenarios: class-wise non-iid and Dirichlet non-iid. We conduct extensive comparative analysis under these two settings throughout our experiments. In contrast, FLASH only considers the Dirichlet non-iid setting.
>
> We are impressed by the superior performance of FLASH and hold it in high regard for its contributions. We have included it in our revised paper.

---

### Author Response · Authors · 2023-11-21
**New Theoretical Analyses of FedP3 and LDP-FedP3**

We provide new analyses exploring the theoretical benefits of FedP3 and its locally differential-private variant, LDP-FedP3, focusing on aspects such as convergence, communication costs, and privacy. The primary updates are summarized as follows:

- **Global Pruning and Personalized Model Aggregation Sketches**: Definitions 1 and 2 provide detailed descriptions of the global pruning sketch ($\mathbf{P}$) and the personalized model aggregation sketch ($\mathbf{S}$), respectively.

- **Convergence and Communication Cost of FedP3**: Theorem 1 delivers a thorough convergence analysis for FedP3, indicating its communication cost as $\mathcal{O}(d/\epsilon^2)$. This denotes an improvement over unpruned methods by a factor of $\mathcal{O}(n/\epsilon)$ for large $n$, beneficial for scalable FL. Moreover, leveraging the same gradient upper bound from the smoothness assumption (Lemma 1) for baselines, we see a communication cost reduction by a factor of $\mathcal{O}(d/n)$ for free, suggesting the possibility of further theoretical advancements in communication efficiency.

 - **Introduction of LDP-FedP3 and Its Theoretical Analysis**: In Algorithm 5, LDP-FedP3, a version of FedP3 incorporating local differential privacy, is introduced. Theorem 2 elaborates on its privacy guarantee and convergence analysis. We also contrast its total communication cost with existing methods, as depicted in Table 5. Notably, while traditional methods show linear growth in communication complexity with increasing $n$, our approach maintains consistent efficiency. Compared to baselines with akin communication complexity structures, our method significantly surpasses SDM-DSGD and equates the best complexity of standard DCGD as reported in LDP-CGD.

- **Convergence Analysis of Biased Global Pruning**: Appendix C.4 discusses the convergence analysis of biased global pruning. The total iteration complexity for attaining an $\epsilon$-stationary point is $\mathcal{O}(1/\epsilon^2)$, providing crucial insights into the analysis of biased gradient estimators.

---

### Meta-Review · Area_Chair_SBTX · 2023-12-21

**Metareview:**

The paper introduces FedP3, a Federated Personalized and Privacy-friendly Network Pruning Framework, designed to address model heterogeneity effectively. The reviewers think that the paper explores an interesting idea that has practical significance, i.e., personalized network pruning at the client and server sides. FedP3 can adapt established techniques to specific instances, offering a practical solution.

However, the are some concerns that have remained. For example, the communication cost gain is not clear enough (even after rebuttal). The convergence analysis (Appendix C.2) in rebuttal shows that the method needs more rounds (despite reduced message size in each round). Moreover, the experimental results use the same number of rounds for each method, which may be an unfair comparison, and the paper lacks experimental convergence investigations. I strongly encourage the authors to incorporate all the great comments from the reviewers (most of whom are experts in the field).

All in all, I am in favor of accepting this paper despite the fact that it appears as a borderline paper given the scores.

**Justification For Why Not Higher Score:**

See above

**Justification For Why Not Lower Score:**

See above

---

### Decision · Program_Chairs · 2024-01-16

Accept (poster)